# De novo lipogenesis fuels adipocyte autophagosome and lysosome membrane dynamics

Leslie A. Rowland [1] ✉, Adilson Guilherme [1], Felipe Henriques [1], Chloe DiMarzio[1], Sean Munroe[1], Nicole Wetoska[1], Mark Kelly [1], Keith Reddig[2], Gregory Hendricks[2], Meixia Pan[3], Xianlin Han [3,4], Olga R. Ilkayeva [5], Christopher B. Newgard [5] & Michael P. Czech [1] ✉

Adipocytes robustly synthesize fatty acids (FA) from carbohydrate through the de novo lipogenesis (DNL) pathway, yet surprisingly DNL contributes little to their abundant triglyceride stored in lipid droplets. This conundrum raises the hypothesis that adipocyte DNL instead enables membrane expansions to occur in processes like autophagy, which requires an abundant supply of phospholipids. We report here that adipocyte Fasn deficiency in vitro and in vivo markedly impairs autophagy, evident by autophagosome accumulation and severely compromised degradation of the autophagic substrate p62. Our data indicate the impairment occurs at the level of autophagosome-lysosome fusion, and indeed, loss of Fasn decreases certain membrane phosphoinositides necessary for autophagosome and lysosome maturation and fusion. Autophagy dependence on FA produced by Fasn is not fully alleviated by exogenous FA in cultured adipocytes, and interestingly, imaging studies reveal that Fasn colocalizes with nascent autophagosomes. Together, our studies identify DNL as a critical source of FAs to fuel autophagosome and lysosome maturation and fusion in adipocytes.

Adipose tissues are major regulators of whole-body metabolism, and disruptions to adipose tissue function can drastically alter overall health[1]. White adipose tissue (WAT) sequesters energy in the form of triglycerides (TGs), which are derived from two main sources of lipid: exogenous circulating fatty acids and endogenously synthesized fatty acids arising from de novo lipogenesis (DNL). Through DNL, fatty acids are synthesized from acetyl-CoA derived from carbohydrates, amino acids, and other sources. Specifically, acetyl-CoA is converted to malonyl-CoA by acetyl-CoA carboxylase (ACC), and these substrates combine to form palmitate through the activities of fatty acid synthase (Fasn)[2]. Palmitate is in turn the precursor for synthesis of many other fatty acid types. In adipose tissue, DNL is thought to serve mostly for storing excess energy from carbohydrates, amino acids and other carbon sources as fatty acids, which become esterified and sequestered into TGs[3]. This TG storage function of WAT is hypothesized to enhance whole body metabolic health by enhancing glucose disposal and sequestering toxic lipids away from metabolic tissues such as liver and skeletal muscle[4–7].

Although the above pathways are well established, adipocyte DNL surprisingly accounts for <2% of adipose TG content; instead, most adipocyte TGs are derived from circulating TGs found in lipoproteins[4,8]. Supporting the notion that DNL contributes minimally

[1]Program in Molecular Medicine, University of Massachusetts Chan Medical School, Worcester, MA 01605, USA. [2]Department of Radiology, University of Massachusetts Chan Medical School, Worcester, MA 01605, USA. [3]Barshop Institute for Longevity and Aging Studies, University of Texas Health Science Center at San Antonio, San Antonio, TX 78229, USA. [4]Department of Medicine, University of Texas Health Science Center at San Antonio, San Antonio, TX 78229, USA. [5]Duke Molecular Physiology Institute, Department of Medicine, Division of Endocrinology, Metabolism and Nutrition, Duke University School of Medicine, Durham, NC 27710, USA. ✉e-mail: Leslie.rowland@umassmed.edu; Michael.Czech@umassmed.edu

to TG in fat stores, loss of adipocyte Fasn, the rate-limiting enzyme of DNL, has little effect on total lipid accumulation in adipose tissues in mice under normal feeding conditions[9]. Instead, adipocyte specific-Fasn knockout mice exhibit browning of white adipose tissue and improved glucose homeostasis[9,10]. These data raise the idea that DNL and Fasn may be playing important roles in adipocytes beyond merely being a source of fatty acids for storage in lipid droplets. In fact, in a myriad of cell types, Fasn is often necessary for cell survival or proper cellular function. Some of these functions include maintenance of native sarcoplasmic reticulum membrane composition in muscle and plasma membrane cholesterol in macrophages and retina, Schwann cell myelination, and production of ligands for PPAR transcription factors[9,11–14]. The main cellular role of Fasn in adipocytes, however, has remained a mystery.

The studies described here identify a critical role for Fasn in adipocyte autophagy. Autophagy is a degradative process in which cellular components are broken down and recycled by lysosomes[15]. Autophagy can promote survival under stress conditions, such as a means of recycling nutrients in response to starvation, but also serves homeostatic roles in the clearance of damaged or unneeded cellular components[15,16]. During autophagy, cellular contents destined for degradation are encompassed and sequestered by autophagosomes, double-membrane vesicles that ultimately fuse with lysosomes. Autophagosomes form on-demand, and it's estimated their growth requires an astonishing ~4000 phospholipids per second, but the source of these membrane lipids has largely remained elusive[17]. Our results presented here demonstrate that fatty acids produced by Fasn are a critical source of lipids for these remarkable autophagosome dynamics. This conclusion is consistent with studies in both yeast and mammalian cells that have suggested a link between de novo lipid synthesis and the growing phagophore[18–21]. Indeed, we show that adipocytes require localized fatty acid synthesis for efficient autophagy, as exogenous fatty acid supplementation does not fully rescue the defects observed in Fasn-deficient cells. These new data demonstrate a key role of adipocyte DNL in autophagy, as opposed to its relatively minor contribution to adipocyte TG storage.

## Results

### Fasn-deficient adipocytes display impaired autophagy

In contrast to adipocytes in vivo, cultured adipocytes that are differentiated from progenitor cells (preadipocytes) in vitro rely almost exclusively on DNL for TG synthesis and lipid droplet formation due to the low fatty acid content of cell culture media. Therefore, we reasoned that investigating Fasn function might be most productively achieved in cultured adipocytes. Although Fasn is required for adipocyte differentiation, Fasn-deficient adipocytes can be obtained from preadipocytes derived from Fasn[Fl/Fl] mice harboring adiponectin promoter-driven Cre recombinase. In such cells, Fasn deletion is delayed until after onset of adipocyte differentiation, enabling adipocyte formation. Indeed, preadipocytes from Fasn[Fl/Fl] and Fasn[Fl/Fl]; Adiponectin-Cre[+] (cAdFasnKO) mice differentiate competently into adipocytes (Fig. 1a). However, whereas cAdFasnKO adipocytes initially accumulate lipid droplets in a manner similar to Fasn-expressing control cells, they lose most of their lipids by Day 7 post-differentiation (Fig. 1a), due to an almost complete lack of fatty acid synthesis (Supplementary Fig. 1A). Interestingly, this loss of lipids does not appear to be accompanied by a de-differentiation of the cells, as they maintain their rounded adipocyte morphology and express genes known to be abundant in adipocytes (Fig. 1b).

Imaging of Fasn-deficient adipocytes by electron microscopy revealed a striking abundance of autophagic vesicles, characterized by a double membrane surrounding cytoplasmic content/organelles in various states of degradation (Fig. 1c). Accumulation of autophagic vesicles correlated as expected with a clear increase in levels of the autophagosome marker LC3 (Fig. 1d, e). We also assessed levels of p62,

a mediator of selective autophagy and an autophagy substrate. Levels of this protein were strongly increased (~6-fold) in cAdFasnKO adipocytes (Fig. 1d, e), which could not be fully attributed to a much smaller increase (~2-fold) in levels of the p62/Sqstm1 transcript (Fig. 1f), suggesting a potential stabilization of p62 or a decrease in its degradation. Immunofluorescence labeling also showed strong increases in LC3 and p62 protein in cAdFasnKO adipocytes (Supplementary Fig. 1B). Gene expression analysis showed that Becn1 and Mitf transcripts were slightly elevated in KO adipocytes, whereas expression of other autophagy-related genes was unaffected (Fig. 1f).

After proteolytic processing from a common precursor (proLC3), the LC3 protein exists in two forms in the cell: LC3I and LC3II. LC3II differs from LC3I in that upon autophagy activation, the LC3I protein becomes conjugated to phosphatidylethanolamine, forming LC3II. LC3II is then anchored into the autophagosome membrane and thus serves as a reliable marker of both growing and completed autophagosomes. Because an increase in LC3II could indicate an increase in autophagosome formation but also a decrease in autophagosome degradation, measuring changes in LC3II in response to stimuli that activate or block autophagy is necessary to adequately assess autophagic flux[22]. The increases in LC3 and p62 proteins in cAdFasnKO adipocytes suggesting altered autophagy flux prompted us to investigate further with an LC3II turnover assay. Adipocytes were treated with HBSS to activate autophagy and chloroquine (CQ) to block autophagic degradation. Shown in Fig. 1g and quantified in Fig. S1C, activation of autophagy with HBSS had no effect on LC3II levels in either control or KO adipocytes in the absence of CQ. Blockade of autophagic degradation by CQ, however, increased LC3II in WT and KO adipocytes (Fig. 1g and Supplementary Fig. 1C). This large increase in LC3II accumulation with CQ but not with HBSS indicates these adipocytes likely already have a high rate of autophagic flux under basal conditions. Since LC3II levels differ between Fasn[Fl/Fl] and cAdFasnKO adipocytes under basal conditions (Fig. 1d, g), it is necessary to calculate the ratios of LC3II in the presence of HBSS and CQ to accurately determine flux[22]. Surprisingly, the LC3II synthesis ratio, a measure of autophagosome formation rate, is not significantly reduced by FasnKO (Fig. 1h). However, calculation of the LC3II degradation ratio, a measure of autophagic degradation, showed a significant reduction by FasnKO (Fig. 1i). These data indicate that autophagic flux is impaired in cAdFasnKO adipocytes due to reductions in autophagic degradation and not autophagy activation.

Since p62 is an autophagy substrate, p62 degradation can be another useful way to assess autophagy flux in cells. Under conditions in which autophagy is inhibited, p62 degradation is impaired, and accumulated p62 condenses to form detergent-insoluble aggregates[22]. We assessed the presence of these insoluble aggregates by fractionating adipocyte protein lysates into Triton X-100-soluble and Triton X-100- insoluble forms. In the soluble fraction of Fasn KO cells, p62 is significantly increased similar to the data shown in Fig. 1d, in concert with a massive accumulation of p62 in the insoluble fraction (Fig. 1j and k). These data further confirm autophagic degradation is strongly impaired in cAdFasnKO adipocytes. Furthermore, the abundance of enclosed autophagosomes (Fig. 1c) as well as the increase in LC3 lipidation in response to autophagy modulation (Fig. 1g) in Fasn deficient adipocytes indicate they are not defective in early stages of autophagy but rather have compromised fusion with the lysosome leading to impaired degradation.

### Protein malonylation is increased in cAdFasnKO adipocytes

While loss of Fasn eliminates the production of palmitate via DNL, it can also affect levels of earlier pathway intermediates, such as acetyl-CoA and malonyl-CoA[23,24] (Fig. 2a). The role of malonyl-CoA in autophagy has not been determined, whereas acetyl-CoA is a

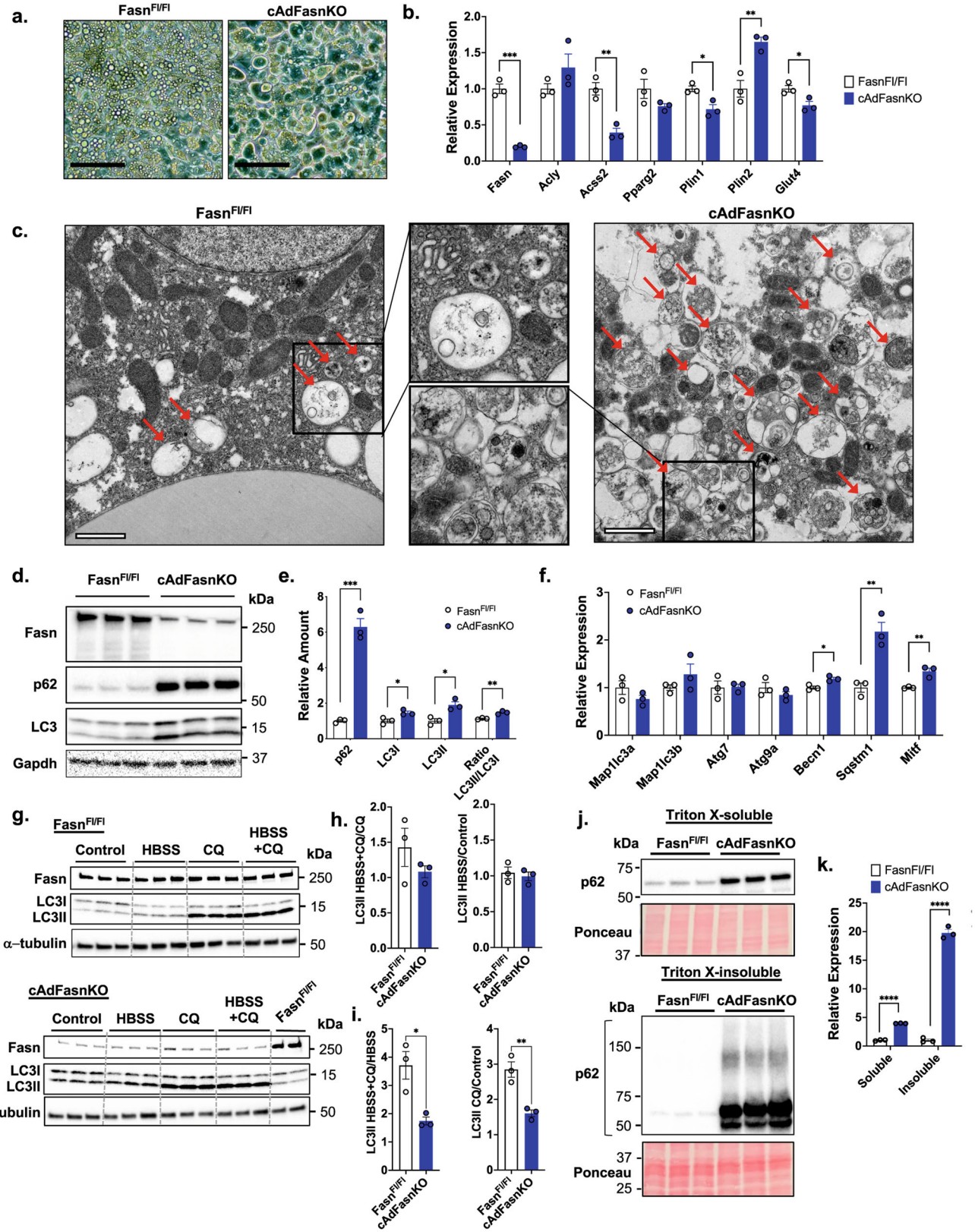

well established negative regulator of autophagy. Reductions in cytosolic acetyl-CoA activate, while increases in acetyl-CoA inhibit autophagy[25]. For these reasons, it was important to establish whether the impairment of autophagy in cAdFasnKO adipocytes might be linked to reduced palmitate production or altered levels of acetyl-CoA or malonyl-CoA (Fig. 2a).

To determine whether acetyl-CoA or malonyl-CoA levels were affected by Fasn deletion, LC-MS/MS was performed on cell extracts. Surprisingly, total acetyl-CoA levels were significantly reduced, indicating elevated acetyl-CoA is unlikely to play a role in the autophagy inhibition observed in cAdFasnKO adipocytes. In contrast, malonyl-CoA was slightly but not significantly elevated in

**Fig. 1 | Impaired autophagy in adipocytes deficient in Fasn. a** Light microscopy of in vitro differentiated FasnFl/Fl and cAdFasnKO primary adipocytes. Scale bar = 50 μm. **b** qPCR of DNL genes and adipocyte markers. *n* = 3 samples, similar data obtained in 2 independent experiments. **c** Electron micrographs of same adipocytes as in (**a**). Arrows indicate autophagic vesicles. Scale bar = 1 μm. **d** Western blot and (**e**) quantification of autophagic markers of FasnFl/Fl and cAdFasnKO adipocytes. Proteins normalized to Gapdh. *n* = 3 samples, similar data obtained in at least 2 independent experiments. **f** qPCR of autophagy-related genes in FasnFl/Fl and cAdFasnKO primary adipocytes. *Map1lc3a* = LC3A, *Map1lc3b* = LC3B, *Sqstm1* = p62. *n* = 3 samples, similar data obtained in at least 2 independent experiments. **g** Western blots of LC3II turnover assay in FasnFl/Fl and cAdFasnKO adipocytes. Cells were starved (in HBSS) and/or treated with CQ for 4 h. HBSS = Hanks' balanced salt solution, CQ = 50 μM chloroquine. **h** Synthesis ratio of LC3II calculated as the ratio of LC3II with HBSS to the no HBSS condition. The ratio was calculated with and without autophagy inhibition with CQ as indicated. *n* = 3 samples, similar data obtained in at least 2 independent experiments. **i** Degradation ratio of LC3II calculated as the ratio of LC3II in the presence of CQ to no CQ treatment. The ratio was calculated with and without autophagy induction with HBSS as indicated. *n* = 3 samples, similar data obtained in at least 2 independent experiments. **j** Western blots for p62 of Triton X-100-soluble and -insoluble protein fractions. Ponceau S staining provided for loading control. **k** Quantification of blots in (**j**). *n* = 3 samples, similar data obtained in at least 2 independent experiments. All data are means ± SE. Two-tailed *t* tests: *P* values from left to right (**b**) 0.000268, 0.215699, 0.00448, 0.151227, 0.020024, 0.008245, 0.032821, (**e**) 0.000359, 0.029211, 0.015753, 0.004466, (**f**) 0.213048, 0.27397, 0.910372, 0.293104, 0.014782, 0.005682, 0.007374, (**h**) 0.2892, 0.6414, (**i**) 0.0182, 0.0065, (**k**) 0.000002, 0.000008. *<0.05, **<0.01, ***<0.001, ****<0.0001. Source data are provided as a Source Data file.

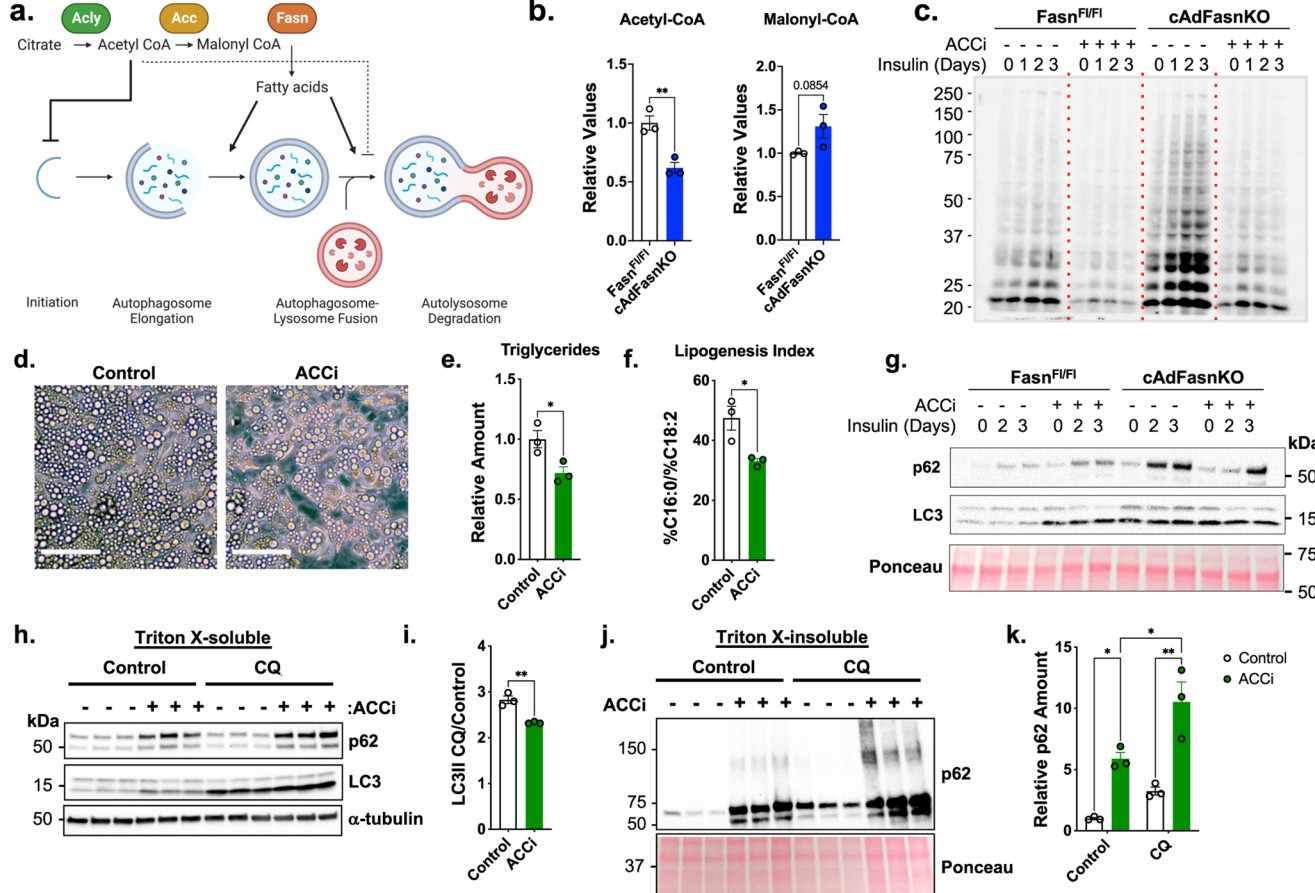

**Fig. 2 | DNL metabolites acetyl-CoA and malonyl-CoA do not mediate the impaired autophagy in cultured FasnKO adipocytes. a** Diagram of de novo lipogenesis and its proposed role(s) in autophagy. Figure created with BioRender. **b** Relative acetyl-CoA and malonyl-CoA levels in FasnFl/Fl and cAdFasnKO adipocytes determined by LC-MS/MS. *n* = 3 samples. Two-tailed *t* test: *P* value (Acetyl-CoA) = 0.0075, **<0.01. **c** Western blot of malonylated proteins in FasnFl/Fl and cAdFasnKO adipocytes in the presence of 15 μM ND-630 (ACC inhibitor, ACCi) for 72 h and 1 μM insulin for the indicated time points. **d** Light microscopy of in vitro differentiated adipocytes treated with and without 10 μM ACCi for 48 h. Scale bar = 50 μm. **e** Relative triglyceride content and (**f**) Lipogenesis index calculated as the ratio of the percentage of C16:0 to the percentage of C18:2 in triglycerides in control and ACCi-treated adipocytes. *n* = 3 samples. Two-tailed *t* test: *P* values = (**E**) 0.0336, (**f**) 0.0245, *<0.05. **g** Western blot of soluble p62 and LC3 in FasnFl/Fl and cAdFasnKO adipocytes treated with ACCi and insulin as shown in (**c**). **h** Western blot for p62 and LC3 of Triton X-100-soluble protein fractions from adipocytes treated with 10 μM ACCi for 48 h with and without 4-h treatment with 50 μM CQ. **i** LC3II degradation ratio calculated from Fig. 2H. *n* = 3 samples, similar data obtained in at least 2 independent experiments. Two-tailed *t* test: *P* Value = 0.0038, **<0.01. **j** Western blot for p62 of Triton X-100-insoluble protein fractions from adipocytes treated with 10 μM ACCi for 48 h with and without 4-h treatment with 50 μM CQ. **k** Quantification of insoluble p62 from 2 J. Ponceau provided as loading control. *n* = 3 samples, similar data obtained in at least 2 independent experiments. 2-way ANOVA with Tukey post hoc: *P* values: Main effects: CQ = 0.0042, ACCi = 0.0001, Interaction = 0.2040, Control vs. ACCi = 0.0173, CQ control vs. CQ ACCi = 0.0016, ACCi control vs. CQ ACCi = 0.0225, *<0.05, **<0.01. All data are means ± SE. Source data are provided as a Source Data file.

cAdFasnKO adipocytes (Fig. 2b), and immunoblot analysis of total malonylated proteins demonstrated a dramatic increase in cAdFasnKO adipocytes (Fig. 2c). This discrepancy between malonyl-CoA levels and malonylated proteins may suggest that protein malonylation serves as a sink for excess malonyl-CoA. These data suggest elevated malonyl-CoA or protein malonylation could play a role in the autophagy inhibition observed in cAdFasnKO adipocytes.

## Protein malonylation in cAdFasnKO adipocytes does not mediate autophagy inhibition

To determine whether protein malonylation or malonyl-CoA impairs autophagy, we used firsocostat/ND-630, a pharmacological inhibitor of ACC1/2, the enzyme responsible for malonyl-CoA production and immediately upstream of Fasn (Fig. 2a). Inhibition of ACC in fully differentiated cultured adipocytes caused a slight diminution of lipid droplets as observed by light microscopy (Fig. 2d), accompanied by a reduction in cellular triglycerides (Fig. 2e). Quantification of the lipogenesis index, the ratio of C16:0 to C18:2, in triglycerides and a measure of DNL[26,27], indicated that DNL was effectively inhibited by ND-630 treatment (Fig. 2f). ND-630 treatment also successfully reduced protein malonylation in control cells and returned protein malonylation levels to control levels in cAdFasnKO adipocytes (Fig. 2c). Together, these data demonstrate that pharmacological inhibition of ACC in fully differentiated adipocytes effectively reduces protein malonylation and DNL.

Western blotting for autophagy markers LC3 and p62 showed that ACC inhibition alone is sufficient to increase LC3II and p62 content, while ACC inhibition in cAdFasnKO adipocytes does not restore LC3II or p62 to control levels, indicating increased protein malonylation does not contribute to autophagy impairment (Fig. 2g). Since we noticed an effect of ND-630 alone on autophagy markers, we measured autophagy flux in the ND-630-treated adipocytes in the presence of CQ. Similar to the findings with cAdFasnKO adipocytes, ACC inhibition significantly reduces autophagy flux measured by the LC3II degradation ratio (Fig. 2h, i, and Supplementary Fig. 2A). ACC inhibition also significantly increases both soluble and insoluble p62 levels (Fig. 2h and j, quantified in Supplementary Fig. 2B and K, respectively). Combined, these data show that like cAdFasnKO, ACC inhibition in adipocytes significantly impairs autophagic degradation, providing independent evidence that inhibition of fatty acid synthesis is sufficient to impair autophagy.

## Fatty acids produced by Fasn are essential for autophagy

If fatty acid synthesis via Fasn is required for autophagy, we hypothesized that the provision of exogenous fatty acids would restore autophagy function in cAdFasnKO adipocytes. Supplementation of cAdFasnKO adipocytes with a mixture of palmitate and oleate indeed restored soluble p62 levels, though insoluble p62 levels were only partially rescued (Fig. 3a, b). Likewise, LC3II was not restored by the addition of fatty acids to cAdFasnKO adipocytes (Fig. 3a and c). Similar results were obtained when fatty acids were provided for a longer period and throughout the differentiation process, where lipid droplets were plentiful in KO adipocytes (Supplementary Fig. 3A–D). Thus, the autophagy impairment in cAdFasnKO adipocytes could not be ascribed simply to a lack of adipocyte lipid content. Importantly, p62/Sqstm1 mRNA levels were not reduced by fatty acids (Fig. 3d), though other autophagy gene expression changes as a result of cAdFasnKO were restored by fatty acids (Fig. 3e). Immunofluorescent labeling similarly showed a partial reduction of p62 levels in cAdFasnKO adipocytes by fatty acids (Fig. 3f). These data suggest that the partial rescue of p62 protein levels was a direct consequence of fatty acids restoring p62 degradation and thus autophagy flux in FasnKO adipocytes.

Because exogenous fatty acids and lipid droplet restoration were not sufficient to fully rescue autophagic flux in knockout adipocytes, we hypothesized that local fatty acid synthesis is required for proper autophagosome dynamics. To test this, we performed immunofluorescence colocalization analyses to determine if Fasn colocalized with the autophagosome marker, LC3B, shown in Fig. 3g. As expected, LC3B puncta formation was increased by starvation in HBSS (Fig. 3g). Fasn immunolabeling showed diffuse cytoplasmic staining but also appeared in punctate structures, which were increased by autophagy-activating conditions; i.e. starvation. Interestingly, when autophagy

was activated by starvation, the amount of LC3B colocalized with Fasn significantly increased (Fig. 3h). Importantly, increasing LC3B puncta formation by autophagy inhibition (CQ) did not increase Fasn/LC3B colocalization; whereas, autophagy activation under these conditions did increase colocalization (Fig. 3i). These data indicate Fasn localizes with nascent autophagosomes and likely contributes to the growing autophagosome membrane. Together, these data are consistent with the hypothesis that endogenous fatty acid synthesis directly supplies lipids to autophagosome membranes and is required for effective autophagic flux.

## Impaired autophagy in adipocytes of cAdFasnKO mice

We next asked whether the autophagy defect in Fasn-deficient adipocytes could be observed in vivo. For these experiments, we used tamoxifen-inducible, adipocyte specific-FasnKO (iAdFasnKO) mice, as well as the constitutively active Adiponectin-Cre (cAdFasnKO) mice that were used for the previous studies on cultured adipocytes. In subcutaneous white adipose tissue of iAdFasnKO mice, we observed increased p62 and LC3II (Fig. 4a), consistent with our in vitro experiments. Also observed was the characteristic beiging of iAdFasnKO mice evident by increased Ucp1 expression. Similar results were found after mice were fasted, an autophagy-activating condition (Fig. 4a). These changes in autophagic markers occurred specifically in adipocytes, shown by fractionation of adipose tissue (Fig. 4b) and immunohistochemical staining of p62 in WAT (Fig. 4c). Increased p62 protein was observed in all adipose tissues but was not due to increased Sqstm1 transcription (Fig. 4d and e, respectively). An ex vivo autophagy flux assay assessing LC3II and p62 changes in the presence of CQ showed significantly increased LC3II and p62 in both basal and CQ-treated conditions of cAdFasnKO WAT (Fig. 4f, g). Calculation of the degradation ratio for LC3II indicated a tendency for a reduced autophagosome degradation in KO explants (Fig. 4h), although there was a significant reduction in p62 degradation (Fig. 4h). These results suggest an impairment in autophagosome degradation in cAdFasnKO adipose tissue in vivo, consistent with our findings in cultured cAdFasnKO adipocytes in vitro.

Based on the partial restoration of autophagy flux by fatty acid supplementation in vitro, mice were fed a high fat diet (HFD) rich in saturated fatty acids to determine if this would restore autophagy flux in vivo. Shown in Fig. 4i, HFD feeding reduced p62 protein levels and interestingly also blunted Ucp1 protein levels in iAdFasnKO iWAT. Again, these p62/Sqstm1 changes were not due to transcriptional effects (Fig. 4j). Changes in LC3 lipidation were not as clear in this cohort of chow-fed iAdFasnKO mice. We hypothesize these variations are due to the cellular heterogeneity of the adipose tissue in vivo as we see a significant amount of LC3 in the SVF fraction (Fig. 4b). These data in mice reinforce our findings with cAdFasnKO adipocytes in culture indicating Fasn is required for proper autophagy flux in adipocytes, and exogenous fatty acids can only partially restore autophagic function.

## General proteostasis is unaltered by Fasn deficiency

The autophagic degradation system and the ubiquitin-proteasome system are the major pathways for protein degradation and maintenance of proteostasis, thus blockade of either arm can lead to an accumulation of misfolded and damaged proteins. In addition, crosstalk between these systems has been established, illustrated by the fact that impairment of proteasomal degradation can activate autophagy as a compensatory mechanism, though it's not clear if the alternative is true[28,29]. To determine if general proteostasis was altered by Fasn deletion or inhibition, we measured total ubiquitinated protein levels. Inhibition of Fasn in cultured adipocytes using the potent inhibitor, TVB3664, increased soluble p62 and lipidated and unlipidated LC3B, as found with FasnKO adipocytes, though total ubiquitinated proteins were not affected (Supplementary Fig. 4A). Treatment with the

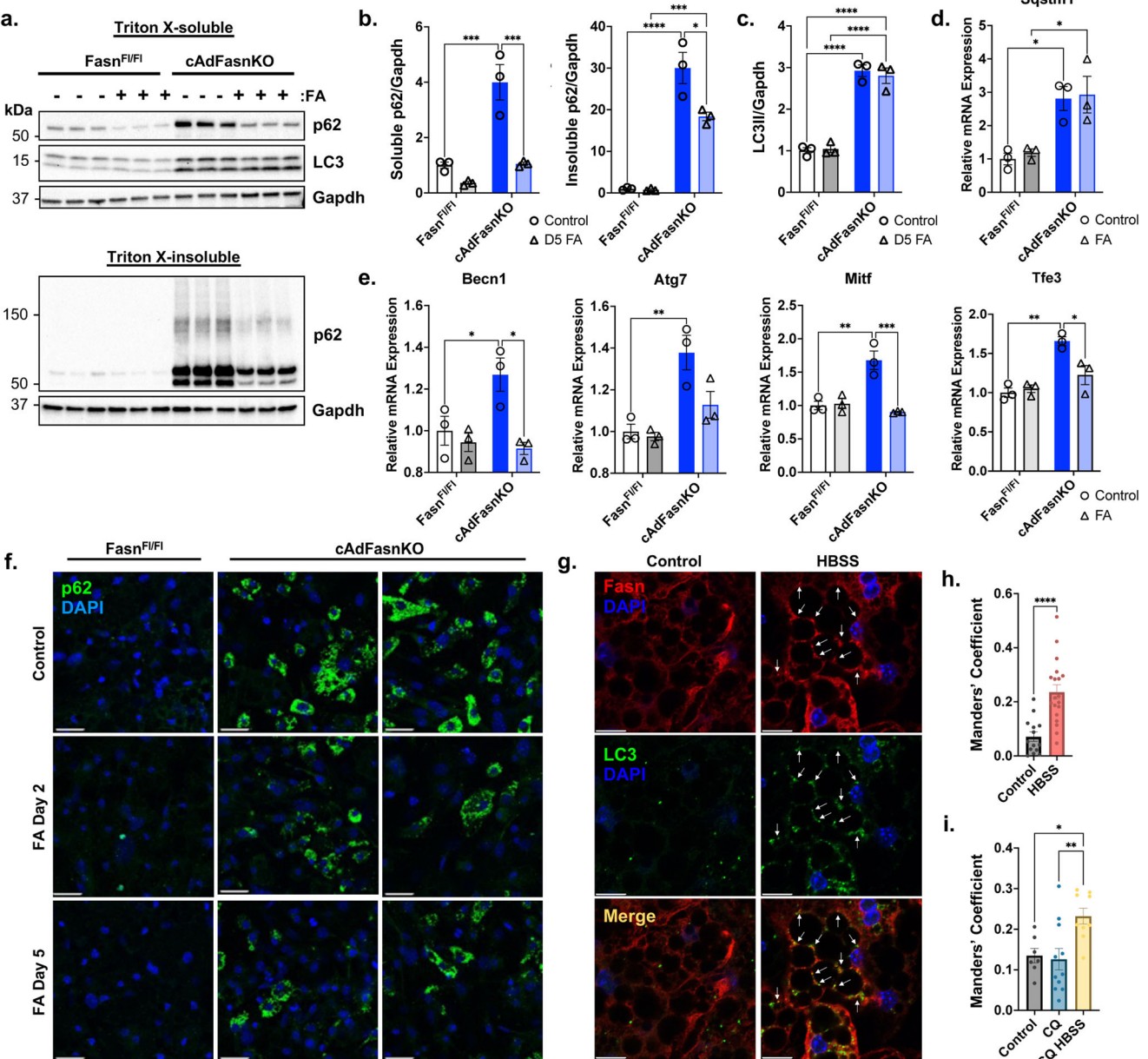

**Fig. 3 | Fatty acids produced by Fasn are essential for autophagy.** Fasn[Fl/Fl] and cAdFasnKO adipocytes were supplemented with 200 µM each of palmitate and oleate for 48 h after they were fully differentiated (day 5). **a** Western blots of Triton X-100-soluble and -insoluble protein fractions. **b** Quantification of soluble and insoluble p62 in Fig. 3A normalized to Gapdh. **c** Quantification of LC3II normalized to Gapdh from Fig. 3A. **d** qPCR of *Sqstm1*/p62 and (**e**) autophagy-related genes in adipocytes supplemented with fatty acids (FAs) throughout differentiation, beginning on day 0. *n* = 3 samples, similar data obtained in at least 2 independent experiments. (**b**−**e**) All data are means ± SE. All datasets (**b**−**e**) analyzed with two-way ANOVA with Tukey post hoc: *P* values = (B, soluble p62) Fasn[Fl/Fl] vs. cAdFasnKO 0.0009, Fasn[Fl/Fl] vs. Fasn[Fl/Fl] FA 0.5464, Fasn[Fl/Fl] FA vs. cAdFasnKO FA 0.4807, cAd-FasnKO vs. cAdFasnKO FA 0.0010, (**b**, insoluble p62) Fasn[Fl/Fl] vs. cAdFasnKO <0.0001, Fasn[Fl/Fl] vs. Fasn[Fl/Fl] FA 0.9998, Fasn[Fl/Fl] FA vs. cAdFasnKO FA 0.0010, cAdFasnKO vs. cAdFasnKO FA 0.0122, (**c**) Fasn[Fl/Fl] vs. cAdFasnKO <0.0001, Fasn[Fl/Fl] vs. Fasn[Fl/Fl] FA 0.9948, Fasn[Fl/Fl] FA vs. cAdFasnKO FA < 0.0001, cAdFasnKO vs. cAdFasnKO FA 0.9092, (**d**) Fasn[Fl/Fl] vs. cAdFasnKO 0.0239, Fasn[Fl/Fl] vs. Fasn[Fl/Fl] FA 0.9784, Fasn[Fl/Fl] FA vs. cAdFasnKO FA 0.0298, cAdFasnKO vs. cAdFasnKO FA 0.9952, (**e**, Becn1) Fasn[Fl/Fl] vs. cAdFasnKO 0.0491, Fasn[Fl/Fl] vs. Fasn[Fl/Fl] FA 0.9067, Fasn[Fl/Fl] FA vs. cAdFasnKO FA 0.9845, cAdFasnKO vs. cAdFasnKO FA 0.0123,

(**e**, Atg7) Fasn[Fl/Fl] vs. cAdFasnKO 0.0061, Fasn[Fl/Fl] vs. Fasn[Fl/Fl] FA 0.9898, Fasn[Fl/Fl] FA vs. cAdFasnKO FA 0.2938, cAdFasnKO vs. cAdFasnKO FA 0.0532, (**e**, Mitf) Fasn[Fl/Fl] vs. cAdFasnKO 0.0022, Fasn[Fl/Fl] vs. Fasn[Fl/Fl] FA 0.9958, Fasn[Fl/Fl] FA vs. cAdFasnKO FA 0.7404, cAdFasnKO vs. cAdFasnKO FA 0.0009, (**e**, Tfe3) Fasn[Fl/Fl] vs. cAdFasnKO 0.0014, Fasn[Fl/Fl] vs. Fasn[Fl/Fl] FA 0.9655, Fasn[Fl/Fl] FA vs. cAdFasnKO FA 0.4359, cAd-FasnKO vs. cAdFasnKO FA 0.0177, *<0.05, **<0.01, ***<0.001, ****<0.0001. **f** Immunofluorescence of Fasn[Fl/Fl] and cAdFasnKO adipocytes supplemented with FAs beginning on Day 2 or Day 5 of differentiation and labeled with p62. Scale bar = 50 µm. **g** Immunofluorescence imaging of wild-type adipocytes starved for 2 h in HBSS and double-labeled for LC3B and Fasn. Scale bar = 10 µm. **h** Quantification of colocalization of LC3B and Fasn in Fig. 3F as measured by Manders' coefficient = fraction of LC3B colocalized with Fasn. *n* = 14 (Control) and *n* = 19 (HBSS) images from single Z slices. Data are means ± SE. Two-tailed *t* test, *P* value = <0.0001 ****<0.0001. **i** Manders' coefficient for adipocytes treated with 30 µM CQ for 24 h (CQ condition) or starved in HBSS with 50 µM CQ for 5 h (CQ HBSS condition). *n* = 7 (Control), *n* = 11 (CQ) and *n* = 9 (HBSS) images from single Z slices. Data are means ± SE. One-way ANOVA with Sidak post hoc, *P* values = Control vs. CQ 0.9913, Control vs. CQ HBSS 0.0327, CQ vs. CQ HBSS 0.0076, *<0.05, **<0.01. Source data are provided as a Source Data file.

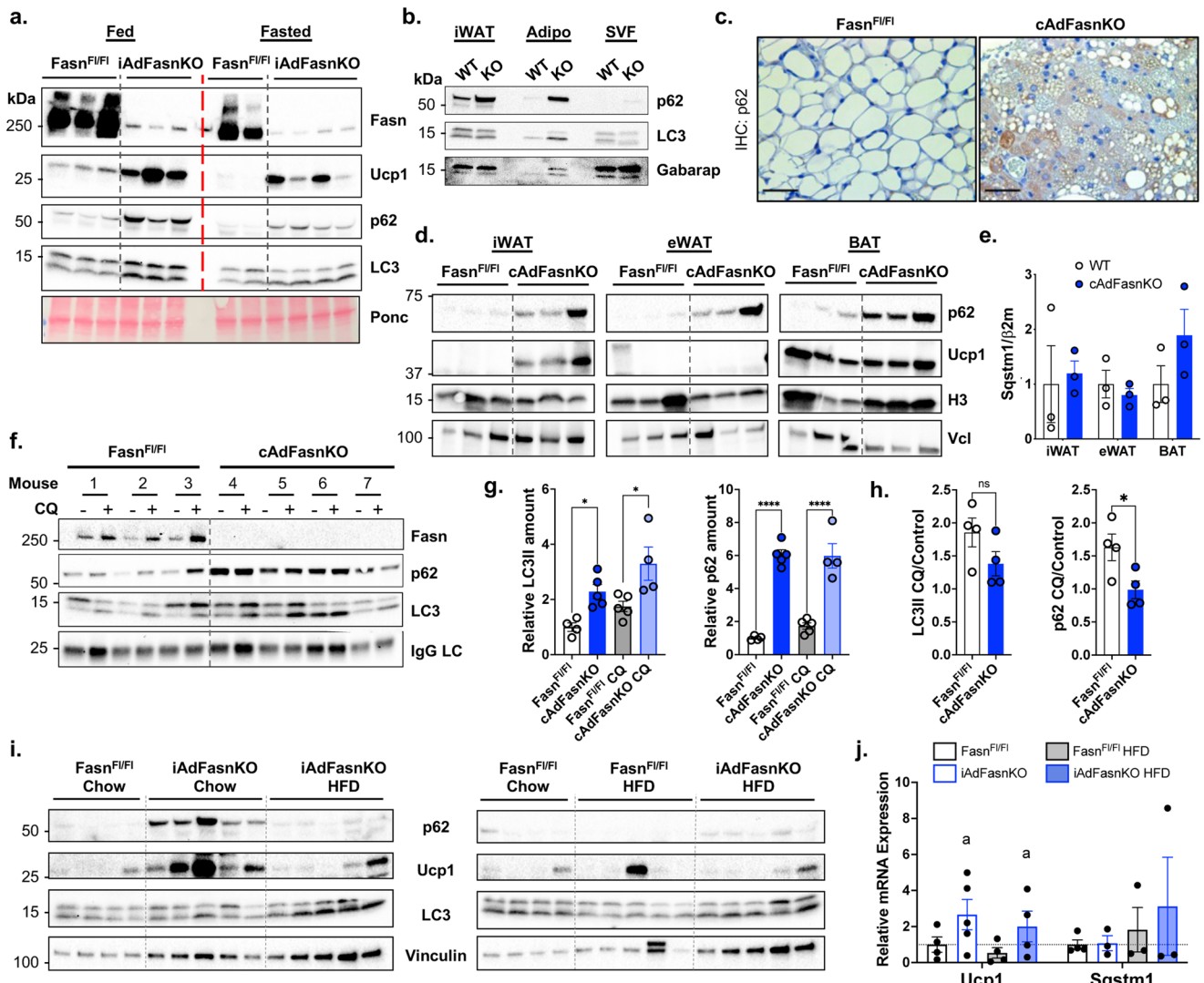

**Fig. 4 | Adipocyte Fasn deficiency impairs autophagy and induces accumulation of p62 protein in vivo. a** Western blot of subcutaneous white adipose tissue (WAT) from Fasn^Fl/Fl and inducible AdFasnKO (iAdFasnKO) mice under fed and 24 h-fasted conditions. Ponceau (Ponc) provided for loading control. $n = 3$ mice (fed, Fasn^Fl/Fl and iAdFasnKO), $n = 2$ mice (fasted, Fasn^Fl/Fl), $n = 4$ mice (fasted, iAdFasnKO). **b** Western blot of pooled ($n = 4$ mice/group) subcutaneous WAT (iWAT) from Fasn^Fl/Fl (WT) and iAdFasnKO (KO) mice separated into adipocyte (Adipo) and stromal vascular fractions (SVF). **c** Immunohistochemical staining of p62 in iWAT of Fasn^Fl/Fl and iAdFasnKO mice; scale bar = 25 μm. Representative image from $n = 4$ mice. **d** p62 Western blot of adipose tissue depots from Fasn^Fl/Fl and cAdFasnKO mice. Histone H3 (H3) and vinculin (Vcl) provided as loading controls. eWAT = epididymal adipose tissue, BAT = brown adipose tissue. $n = 3$ mice. **e** qPCR comparing p62 (*Sqstm1*) mRNA expression across adipose tissue depots. $n = 3$ mice. **f** Ex vivo autophagy flux assay in subcutaneous WAT explants harvested from Fasn^Fl/Fl and cAdFasnKO mice. WAT explants were cultured in serum-free DMEM/F12 media with or without 50 μM CQ for 2 h. Western blot results are shown. **g** Quantification of LC3II and p62 in western blot in Fig. 4F. Proteins were normalized to mouse IgG light chain levels. For LC3II: Mixed-effects model, with Sidak post hoc, $P$ values = Fasn^Fl/Fl vs. cAdFasnKO 0.0238, Fasn^Fl/Fl CQ vs. cAdFasnKO CQ 0.0101, For p62: Mixed-effects model, Sidak post hoc, $P$ values = Fasn^Fl/Fl vs. cAdFasnKO <0.0001, Fasn^Fl/Fl CQ vs. cAdFasnKO CQ < 0.0001, $n = 4$ explants for Fasn^Fl/Fl and cAdFasnKO CQ, $n = 5$ explants for cAdFasnKO and Fasn^Fl/Fl CQ. **h** Degradation ratios of LC3II and p62 were determined by calculating the change in protein amount in the CQ condition divided by the control condition. Two-tailed $t$ tests, $P$ values (left to right) = 0.03215, 0.0276 *<0.05, $n = 4$ mice. **i** Fasn^Fl/Fl and iAdFasnKO mice fed a chow diet or high fat diet. Western blot of subcutaneous WAT. **j** qPCR of *Ucp1* and *Sqstm1* (p62) in samples from 4I. Two-way ANOVA: a = main effect of FasnKO. $n = 4$ mice. All data are means ± SE. Source data are provided as a Source Data file.

proteasomal inhibitor MG132 increased protein ubiquitination similarly in vehicle and Fasn-inhibited adipocytes (Supplementary Fig. 4A). Western blotting of the Triton X-100 insoluble fraction of these lysates showed an increase in insoluble p62 with Fasn inhibition, as expected, accompanied by a mild increase in protein ubiquitination (Supplementary Fig. 4B). MG132 treatment, however, dramatically increased protein ubiquitination in the insoluble fraction in both groups (Supplementary Fig. 4B). The protein p62 contains a ubiquitin binding domain and mediates the turnover of some ubiquitinated proteins, namely insoluble aggregates[30], thus reduced autophagic turnover by Fasn inhibition prevents the degradation of p62-bound ubiquitinated

cargo. Compared to the accumulation of ubiquitinated proteins by proteasomal inhibition (MG132); however, the increase with Fasn inhibition is minor and found only in the insoluble fraction. Moreover, western blotting for total ubiquitinated proteins in WAT lysates from cAdFasnKO mice showed no change (Supplementary Fig. 4C).

Lastly, to ensure proteasomal activity was not altered by FasnKO, we performed a proteasomal activity assay on the WAT samples shown in Supplementary Fig. 4C. We found no differences in activity with any proteasomal substrate (Supplementary Fig. 4D). These findings suggest that while autophagic protein degradation is impaired by Fasn deficiency, general proteostasis is not significantly disrupted.

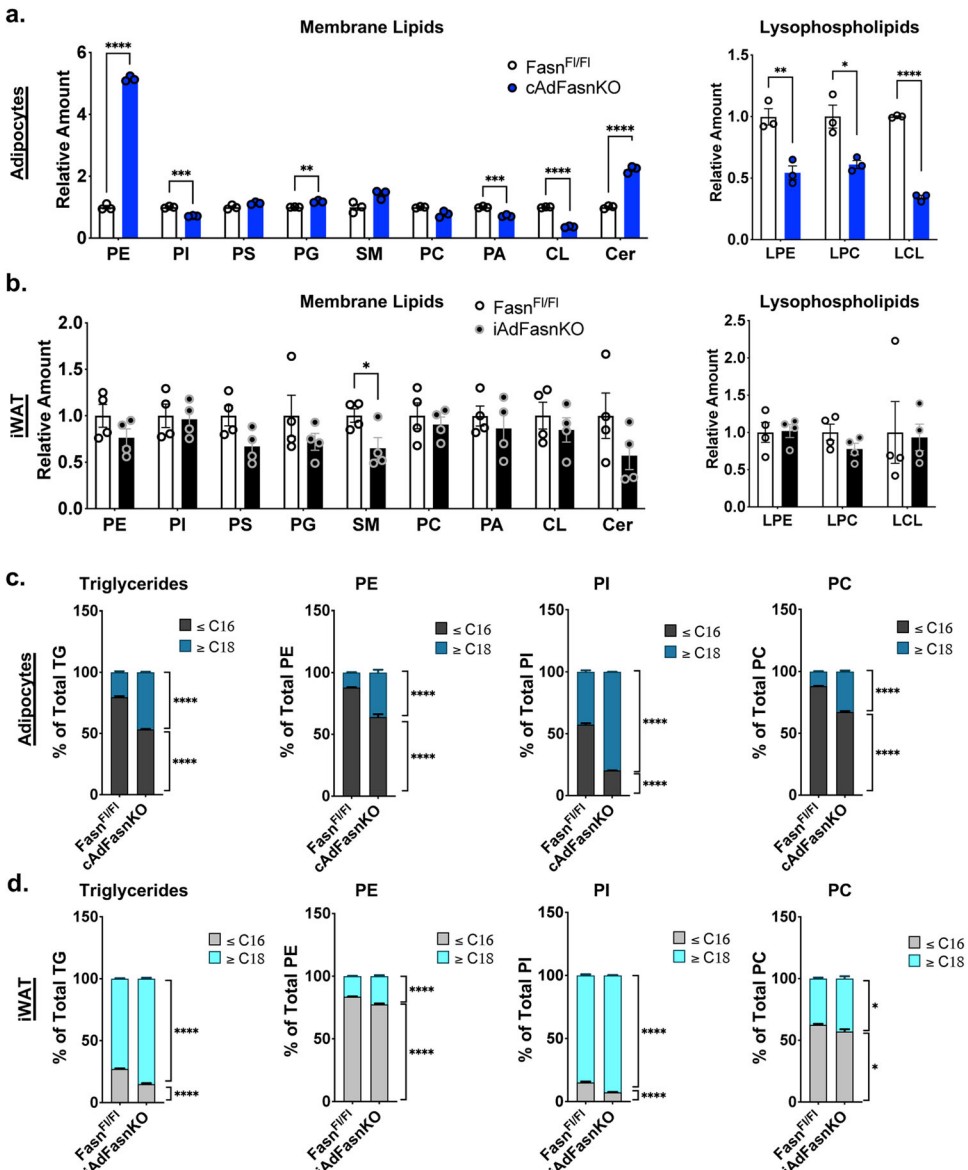

**Fig. 5 | Deletion of Fasn alters the cellular lipidome in vitro and in vivo.** Lipidomics analyses from in vitro differentiated Fasn^Fl/Fl and cAdFasnKO primary adipocytes and subcutaneous WAT from Fasn^Fl/Fl and iAdFasnKO mice. **a** Relative amounts of different lipid classes from in vitro differentiated adipocytes and (**b**) in vivo iWAT. Two-tailed $t$ tests: $P$ Values = (**a**) from left to right <0.00001, 0.000738, 0.061515, 0.004430, 0.061515. 0.054831, 0.001855, 0.000028, 0.00042, 0.013057, 0.016585, <0.00001, (**b**) from left to right 0.178372, 0.820148, 0.057631, 0.284882, 0.041275, 0.580865, 0.50506, 0.457181, 0.184524, 0.90473, 0.141867, 0.885965, *<0.05, **<0.01, ***<0.001, ****<0.0001. Fatty acyl chain composition of different lipid classes in vitro (**c**) and in vivo (**d**). Percent of indicated lipid containing fatty acid chains ≤ 16 carbons vs. lipids with fatty acid chains ≥ 18 carbons. $n$ = 3 samples for in vitro analyses. $n$ = 4 mice for in vivo analyses. All data are means ± SE. Two-way ANOVA with Sidak post hoc, $P$ values = (**c**) from left to right <0.00001/<0.00001, <0.00001/<0.00001, <0.00001/<0.00001, <0.00001/<0.00001, (**d**) from left to right <0.00001/<0.00001, <0.00001/<0.00001, <0.00001/<0.00001, 0.0375/0.0375, *<0.05, ****<0.0001. PE phosphatidylethanolamine, PI phosphatidylinositol, PS phosphatidylserine, PG phosphatidylglycerol, SM sphingomyelin, PC phosphatidylcholine, PA phosphatidic acid, CL cardiolipin, Cer ceramide, LPE lysophosphatidylethanolamine, LPC lysophosphatidylcholine, LCL lysocardiolipin. Source data are provided as a Source Data file.

## Deletion of Fasn alters the cellular lipidome in vitro and in vivo

Given that deletion of Fasn appears to interfere with the degradation of autophagic vesicles but not their formation, Fasn may function to ensure proper membrane lipid composition for autophagosome maturation and lysosome fusion/function. We therefore performed lipidomics analysis on Fasn deficient adipocytes to investigate the effect of Fasn on cellular lipid composition. In vitro, loss of adipocyte Fasn led to significant remodeling of phospholipid composition. Phosphatidylinositol, palmitic acid, cardiolipin, and all measured lysophospholipid species were reduced, while phosphatidylethanolamine, phosphatidylglycerol, and ceramides were increased (Fig. 5a and Supplementary Data 1). In vivo, however, only sphingomyelin was

significantly reduced in adipose tissue (Fig. 5b and Supplementary Data 1).

In mice, the effect of FasnKO on membrane lipids is likely obscured by a constant delivery of fatty acids from the circulation. This prompted us to examine the fatty acyl composition of the lipidome to determine if Fasn is contributing specific fatty acids to each lipid species. We found that Fasn deficiency consistently caused a shift in fatty acyl composition toward longer chain (>18 C) fatty acids in various lipid classes both in vitro (Fig. 5c) and in vivo (Fig. 5d), reflecting a reduction in palmitate (C16:0) synthesis. We also found that loss of Fasn results in shifts in double bond composition in vitro and in vivo, where FasnKO adipocytes contain fatty acyl chains with greater

numbers of double bonds, which arise from essential fatty acids taken up from culture media or dietary intake (Supplementary Fig. 5A and B).

Phosphatidylinositols (PI) are critical for multiple steps of the autophagy pathway, from autophagosome formation to fusion with the lysosome[31,32]. Investigation of the specific phosphatidylinositol species altered by Fasn deficiency in vitro and in vivo showed that PI(16:0-18:2), PI(16:1-20:4), and PI(16:0-20:4) were significantly reduced and PI(18:0-20:4) was significantly increased (Fig. S5C) suggesting these particular PI species may play specific roles in autophagy. Collectively, these data show that loss of Fasn can significantly alter the cellular lipidome and can alter specific phospholipids both in vitro and in vivo.

### Lysosomal activity is impaired by Fasn deficiency

Lysosomal function is also exquisitely sensitive to perturbations in lipid composition[32–34]. Particularly, phosphoinositides and their phosphorylated derivatives regulate the biogenesis and maturation of lysosomes, including their fusion capacity with autophagosomes[34–37]. Since FasnKO showed effects on multiple adipocyte lipids, including PI, we sought to determine whether lysosomal function was also altered by Fasn inhibition or deletion. Lysosomal content is generally estimated by assessing the specific lysosomal membrane protein Lamp1 by western blotting. Shown in Fig. S6A, under nutrient replete conditions, Lamp1 is similar between control and cAdFasnKO adipocytes, despite increased p62 and LC3B. Similarly, inhibition of Fasn by TVB3664 does not affect Lamp1 levels (Supplementary Fig. 6B).

We next examined lysosomal activity using the Magic Red Cathepsin B substrate. Under normal fed conditions lysosomal activity is low and comparable between TVB3664- and vehicle-treated adipocytes (Supplementary Fig. 6C). Upon starvation, vehicle-treated adipocytes display a strong increase in lysosomal activity, indicated by increased fluorescence. However, TVB3664-treated adipocytes show almost no increase in fluorescence, indicating low lysosomal activity (Supplementary Fig. 6C). This defect appears to be at least partially the result of reduced mature Cathepsin B (CTSB) in TVB3664-treated adipocytes, while pro-CTSB appears unaffected (Supplementary Fig. 6D). Lysosomal proteases, such as CTSB, are synthesized as pro-proteins and transported to acidic endosomes then lysosomes, where they are cleaved into their mature form[38]. This difference only in mature CTSB suggests a defect in lysosomal maturation, and together, these data indicate that lysosomal function and membrane dynamics are impaired by Fasn deficiency.

### Fasn inhibition in non-adipocytes impairs autophagy

Fasn and other DNL enzymes are likely universally expressed across mammalian cell types. We thus asked whether inhibition of Fasn in a cell type other than adipocytes would have similar effects on autophagy. In cultured HepG2 cells treated with TVB3664, soluble p62 and lipidated LC3B are indeed significantly increased (Supplementary Fig. 7A–C), as is insoluble p62 (Supplementary Fig. 7D). In addition, an LC3 turnover assay revealed the LC3 degradation ratio is significantly reduced by Fasn inhibition, as found with FasnKO adipocytes, indicating autophagic degradation is compromised (Supplementary Fig. 7E). Notably, calculation of the LC3 synthesis ratio indicated autophagosome synthesis is also significantly reduced by Fasn inhibition (Supplementary Fig. 7F). This is in contrast to FasnKO adipocytes, which do not show a significant impairment in autophagosome synthesis (Fig. 1h). Consistent with our hypothesis, this suggests the large lipid content of adipocytes, even those that are DNL-deficient, can compensate, in part, for the lack of fatty acid synthesis during autophagosome synthesis. In non-adipocytes such as HepG2 cells, where lipid droplets are minimal, the effect of an inhibition of fatty acid synthesis on autophagosome synthesis becomes apparent in addition to the effect on the degradative pathway.

## Discussion

The major finding of this study is the identification of de novo lipogenesis as an essential source of lipids for autophagy in adipocytes. It has previously been appreciated that autophagosome synthesis is an extremely lipid-demanding process, yet the precise source(s) of these lipids has not been fully elucidated[16,17]. Data presented here demonstrate that endogenous de novo fatty acid synthesis via Fasn funnels fatty acids into autophagosomes and lysosomes, and that loss of this pathway disables autophagic degradation via disruption of autophagosome-lysosome fusion and lysosomal function. Bioenergetically, our findings are compatible with the concept that it is more favorable to synthesize new membranes than to reuse old membranes, which require extensive processing such as protein removal[20]. In addition, preferential usage of de novo lipogenesis for membrane lipids rather than depleting preexisting membranes or lipid droplets would appear to confer a benefit to cells, especially during periods of stress. Autophagy serves critical cytoprotective and hormetic roles in virtually all cell types. Thus, our findings answer a fundamental biological question with wide-ranging implications.

Adipose tissue function is paramount to overall health, and the importance of autophagy in adipose tissue function has recently come to light[39,40]. Excessive autophagy induced by adipocyte-selective deletion of the negative autophagy regulator, Rubicon, results in lipodystrophy, glucose intolerance, and fatty liver. Also, downregulation of Rubicon and the associated increase in autophagy has been linked to the age-dependent decline in adipose tissue and metabolic health[41]. Conversely, autophagy inhibition has been associated with improvements in metabolic health. Knockout of Atg7, an early autophagy factor, in adipose tissue blocked autophagy and led to reduced body fat, increased insulin sensitivity, and beiging of white adipose tissue[42]. In addition, several studies have shown that blocking mitophagy, a specialized form of autophagy, prevents the turnover, or "whitening," of beige and brown adipose tissue, leading to beneficial metabolic effects[43–45].

Interestingly, in addition to impaired autophagy, iAdFasnKO and cAdFasnKO mice also exhibit increased white adipose beiging (Fig. 4a, i) and improved glucose tolerance[9,10]. One of the more prominent effects of Fasn knockout on adipocytes was the substantial increase in p62 protein (Fig. 4a, i). In addition to its role in autophagy, p62 serves as a signaling hub that participates in multiple pathways, but most notably, p62 regulates beige and brown adipocyte formation. Knockout of p62 in adipose tissue results in obesity, glucose intolerance, and reduced energy expenditure, effects attributed to reductions in brown and beige adipose tissue thermogenesis[46–48]. Mechanistically, p62 directs phospho-ATF2 to promoters of thermogenic genes and also functions as co-activator of PPARγ[46,48]. It is tempting to speculate that FasnKO-mediated increases in adipose tissue p62 as a result of autophagy dysfunction regulate the beiging and metabolic improvements observed in these mice, but FasnKO in vitro does not upregulate Ucp1 under our experimental conditions. Thus, further investigation is required to determine whether the autophagy impairment in cAdFasnKO mice contributes to development of the beiging phenotype.

A notable finding from these studies was that FasnKO did not impair autophagosome formation in adipocytes, suggested by autophagy flux assays (Fig. 1h) and the presence of apparently fully formed autophagosomes in the electron micrographs (Fig. 1c). Lipid droplets and preexisting membranes have been shown to contribute to autophagosome membranes, and in the absence of de novo lipogenesis, it's likely that multiple membrane lipid sources are utilized[16,49]. Moreover, even though cAdFasnKO adipocytes are largely lipid droplet-depleted, some lipid droplets remain and could serve as lipid sources for autophagosome membranes. Consistent with this hypothesis, inhibition of Fasn in HepG2 cells, a cell line with minimal lipid content, impairs autophagosome synthesis as well as degradation (Supplementary Fig. 7E and F). Current studies are aimed at more

precisely identifying the role of Fasn in autophagy in non-adipocyte cell types where lipid droplets do not accumulate.

Despite the fact that autophagosomes can form in the absence of Fasn, our data strongly suggest that these autophagosomes are not fully functional, i.e., do not fuse with the lysosome for degradation. Autophagic membrane composition must be tightly regulated to ensure proper protein recruitment and membrane dynamics, including autophagosome-lysosome fusion events[31,32,50–52]. Our lipidomics analyses from both cultured adipocytes and adipose tissue showed that Fasn deficiency significantly alters cellular lipid composition, including membrane lipids (Fig. 5). In vitro, knockout of Fasn dramatically alters phospholipid levels, including total PI, PA, PG, PE, among others (Fig. 5a). Phosphatidylinositol, in particular, plays significant roles in autophagy. Its phosphorylated derivatives, PI3P, PI4P, PI(3,5)P$_2$, serve as recognition and docking sites for autophagy effector proteins on both autophagosome and lysosomal membranes[31,32]. Accordingly, reductions in PI synthesis result in impaired autophagy turnover[19]. In addition, alterations to PA content have been shown to affect autophagic flux[53]. In vivo, however, these same changes to total phospholipids were not evident (Fig. 5b) likely due to compensation from uptake of fatty acids from the circulation, though individual species were significantly changed (Supplementary Fig. 5C). Nonetheless, since these lipidomics analyses were performed on whole cells and tissues, we cannot rule out significant alterations to phospholipid profiles at the subcellular level, i.e., in autophagosome or lysosome membranes, where these changes would be impactful. On the other hand, a common theme between the data obtained in Fasn-deficient adipocytes in both cultured cells and in mouse adipose tissue is the significant reduction in C16 and saturated acyl chains across most lipid species, reflecting the reduction in palmitate synthesis (Fig. 5c and Supplementary Fig. 5). Alterations to acyl chain length and saturation can have significant effects on membrane fluidity, thickness, and packing density, which can affect fusion with the lysosome[20,51,54]. Together, these data suggest de novo lipogenesis and Fasn equip the autophagosome membranes with a precise lipid composition that ensures their proper fusion, trafficking, and degradation.

In addition to autophagosome membranes, phospholipid composition is vital to the function and proper maturation of lysosomes[32–34,37]. During autophagy, lysosomes are degraded along with their cargo and must be repopulated to maintain cellular homeostasis. The processes of autophagic lysosome reformation and lysosome repopulation are essential for the maintenance of the lysosomal population and are critically dependent on phosphoinositides[35,36]. Our data showing defective lysosomal function in Fasn-deficient adipocytes (Supplementary Fig. 6) are consistent with the idea that altered lipid, and in particular, PI composition of lysosomal membranes impairs the proper maturation and repopulation of lysosomes.

A prominent finding in this study was that Fasn directly localizes near nascent autophagosomes (Fig. 3f, g). Supportive of this, our results revealed that fatty acids could not fully restore autophagic function of cultured cAdFasnKO adipocytes (Fig. 3a, b), and even fully lipid-replete cAdFasnKO adipose tissue exhibited autophagy impairment (Fig. 4f–h). These results indicate that Fasn may be acting locally to supply fatty acids to growing autophagosomes. Interestingly, Fasn was found to copurify with autophagosomes in a proteomics screen in human cancer cells[55]. In addition, our findings are in line with studies in yeast that showed the fatty acyl CoA synthetase, Faa1, directly localized on growing autophagosome membranes[18]. The phosphatidylinositol (PI) synthesizing enzyme, PI Synthase (PIS) was also shown to localize to sites of autophagosome biogenesis in mammalian cells[19]. The presence of Fasn and lipid synthesizing enzymes on nascent autophagosomes supports the conclusion that one of the primary functions of Fasn is to directly funnel fatty acids into autophagosome membranes.

In summary, our studies provide the first demonstration of a direct contribution of de novo fatty acid synthesis to autophagosome and lysosome membrane dynamics in mammalian cells. Our findings are in line with studies implicating the lipid synthesis enzymes Acc1 and Faa1 in yeast and the fatty acid desaturase, Scd1, in mammalian cells in autophagosome membrane synthesis[18,21,56]. These findings could have far-reaching therapeutic implications as inhibitors of de novo lipogenesis enzymes, particularly ACC and Fasn, are currently in development for the treatment of fatty liver and cancers[57]. Our studies identify an important new function of these enzymes that may impact the usage of these inhibitors. Moreover, dysregulation of autophagy has been implicated in various diseases[58]. Thus, identification of the de novo lipogenesis pathway as a critical component of autophagy opens new opportunities to modulate autophagy and disease progression.

## Methods

### Mice

All mice were raised in standard housing conditions (12 h light/12 h dark cycle, 21-23 °C, 30-35% humidity) and fed a standard chow diet (Prolab IsoPro 3000 #5P76, LabDiet) ad libitum under approval by the University of Massachusetts Chan Medical School Institutional Animal Care Use Committee (IACUC). C57Bl/6 J (WT) mice were obtained from Jackson Laboratory and bred for primary preadipocyte isolation. Fasn$^{Flox/Flox}$ mice were generated as previously described[23] and bred with constitutively expressed Adiponectin-Cre[59] (Jax stock #028020) mice to obtain adipocyte-specific Fasn knockout (KO) mice (cAdFasnKO). For inducible knockout of Fasn in adult mice, Fasn$^{Flox/Flox}$ mice were bred with Adiponectin-Cre-ERT2 mice, as previously described[10]. To induce Fasn knockout, Fasn$^{Flox/Flox}$ or Fasn$^{Flox/Flox}$; Adiponectin-Cre-ERT2$^+$ mice (iAdFasnKO) were administered via i.p. injection 1 mg tamoxifen dissolved in corn oil once per day for 5 days. For high fat diet experiments, mice were fed a 60% kcal from fat diet (Research Diets, D12492i) ad libitum for 4 weeks prior to tamoxifen administration, then remained on the diet for another 12 weeks. Male and female mice were used throughout the study.

### Primary preadipocyte isolation and differentiation

In total, 2–4 week old C57Bl/6 J (WT), Fasn$^{Flox/Flox}$ or cAdFasnKO pups were euthanized and subcutaneous inguinal fat pads were dissected and placed in Hanks' Balanced Salt Solution (HBSS). The fat pads were digested with 1.5 mg/ml Collagenase (Sigma #C6885) with 2% BSA for 40 min at 37 °C with shaking. The digested solution was filtered through a 100 µm filter, centrifuged, and the pellet resuspended in red blood cell lysis buffer. After 2–3 min, the solution was centrifuged again and the pellet resuspended in growth media consisting of DMEM/F12 with 10% fetal bovine serum and 1% (v/v) penicillin/streptomycin. The cells were filtered through a 40 µm filter and grown to confluency. Preadipocytes were used no later than passage 1 for these experiments. To induce differentiation (day 0), preadipocytes were grown to full confluency and cultured in induction medium containing growth media supplemented with 5 µg/mL insulin, 1 µM dexamethasone, 0.5 mM 3-isobutyl-1-methylxanthine, 60 µM indomethacin and 1 µM rosiglitazone. Forty-eight hours later the media was changed to Day 2 media consisting of growth media with 5 µg/mL insulin. The same media was replaced 48 h later (Day 4). Adipocytes were considered fully differentiated at Day 5 and were maintained in regular growth media thereafter. Adipocytes were harvested on Day 6 or Day 7. For fatty acid (FA) supplementation experiments, a 20X stock solution containing 4 mM sodium palmitate, 4 mM oleic acid, and 10% fatty acid free BSA dissolved in DMEM/F12 was prepared. The FAs were conjugated to BSA by heating at 56 °C for ~1 h and the pH was adjusted to ~7.4. The FA mixture was added to the media at 1X concentration (200 µM sodium palmitate, 200 µM oleic acid, 0.5% BSA) on Day 0, Day 2, or Day 5 of differentiation and replaced every other day.

For ACC inhibition experiments, WT primary preadipocytes were differentiated as described above. On day 5, 10 µM firsocostat/ND-630 (S8893, Selleck Chemicals) dissolved in DMSO was added for 48 h. In

experiments with co-treatment with insulin, 1 μM insulin and 15 μM firsocostat were added for 24, 48, or 72 hours beginning on day 5.

For Fasn inhibition experiments, WT primary adipocytes were differentiated and treated with 100 nM TVB3664 (S8563, Selleck Chemicals) or vehicle for 48 hours starting on day 5. For proteostasis experiments, a subset of cells were treated with 20 μM MG132 for 18 hours prior to harvest.

## Lipogenesis assay

Measurement of lipogenesis in cultured cells were performed as described[10,60]. Briefly, differentiated Fasn[Fl/Fl] and cAdFasnKO primary adipocytes were incubated with labeling media containing 0.2% FA-free BSA, 0.5 mM d-glucose, 2 mM sodium pyruvate, 2 mM glutamine and 0.8 mCi/mL $^3$H-[H$_2$O]. 1 mM insulin was added to insulin-stimulated conditions. Adipocytes were incubated at 37 °C with 5% CO2 for 4.5 hours before lipid extraction. $^3$H-[H$_2$O] incorporation into fatty acids was then determined as previously described[10,60,61].

## Autophagy flux Assay/LC3II turnover assay

Cultured adipocytes: For autophagy activation, cultured adipocytes were washed with PBS and incubated in HBSS for 4 h. For autophagy inhibition, adipocytes were cultured in growth media supplemented with 50 μM chloroquine (CQ) (Sigma #6628) for 4 hours. For the combination treatment, 50 μM CQ was added to HBSS and the cells were incubated for 4 h. Control cells were incubated in growth medium. The synthesis ratio, described in[22] was calculated as the change in LC3II in HBSS vs. control or HBSS + CQ vs. CQ for each genotype. Similarly, the degradation ratio was calculated as the change in LC3II in the CQ vs. control or HBSS + CQ vs. HBSS condition.

Adipose Tissue Explants: Freshly dissected subcutaneous adipose tissue depots from individual Fasn[Flox/Flox] or cAdFasnKO male mice were washed in PBS, minced into ~20 mg pieces, and divided into two culture dishes. One dish contained DMEM/F12 and the other contained DMEM/F12 with 50uM CQ, thus each mouse/fat pad was its own control. The explants were incubated at 37 °C with 5% CO$_2$ for 2 hours. After 2 hours, the explants were washed briefly in PBS and protein was isolated as described.

## HepG2 cells

HepG2 cells were grown in DMEM with 10% fetal bovine serum and 1% (v/v) penicillin/streptomycin. Cells were plated in 12-well plates at $2 \times 10^5$ cells/well and grown overnight. The next day, the cells were treated with 100 nM TVB3664 or vehicle and cultured for 48 hours. For the LC3II turnover assay, 18 hours prior to harvest, subsets of cells were treated with 5 μM Rapamycin and/or 50 μM CQ.

## Isolation of adipocytes vs. stromal vascular fraction (SVF)

Subcutaneous white adipose tissue fat pads were dissected and placed in Hanks' Balanced Salt Solution. The fat pads were digested with 1.5 mg/ml Collagenase (Sigma #C6885) for 30 min at 37 °C with shaking. The digested solution was filtered through a 100 μm filter and centrifuged for 5 min at 500 g. The resulting top layer (fat cake) was transferred to a new tube and processed for western blotting as described below. The pellet (SVF) was resuspended in protein homogenization buffer and similarly processed for western blotting.

## Electron microscopy

Fasn[Flox/Flox] and cAdFasnKO preadipocytes were differentiated and on day 7 were fixed by first removing half the plate media and then adding an equal volume of 2.5% glutaraldehyde (v/v) in 1 M Na phosphate buffer (pH 7.2) for 10 min before being transferred to pure 2.5% glutaraldehyde (v/v) in 1 M Na phosphate buffer (pH 7.2) for 1 hour. Next, the cell plates were briefly rinsed (3 × 10 min) in 1 M Na phosphate buffer (pH 7.2) and post-fixed for 1 hr in 1% osmium tetroxide (w/v) in dH$_2$O. Samples were then washed three times with dH$_2$O for 10 mins

and then cells were scraped off the bottom of the wells with a soft plastic spatula, collected in a microfuge tube, and pelleted by centrifugation. Samples were washed three times with dH$_2$O for 10 min and dehydrated through a graded series of ethanol (10, 30, 50, 70, 85, 95% for 20 min each) to three changes of 100% ethanol. Samples were infiltrated first with two changes of 100% Propylene Oxide and then with a 50%/50% propylene oxide/SPI-Pon 812 resin mixture overnight. The following morning the cell pellets were transferred through four changes of fresh SPI-pon 812-Araldite epoxy resin and finally embedded in tubes filled with the same resin and polymerized for 48 hr at 70 °C. The epoxy blocks were then trimmed, and ultrathin sections were cut on a Reichart-Jung ultramicrotome using a diamond knife. The sections were collected and mounted on copper support grids and contrasted with lead citrate and uranyl acetate. The samples were examined on a Philips CM 10 using 100 Kv accelerating voltage. Images were captured using a Gatan TEM CCD camera.

## Real-time quantitative PCR

RNA was isolated from cultured adipocytes or tissue with Trizol following the manufacturer's instructions. cDNA was synthesized from 1μg RNA using Bio-Rad iScript cDNA kit. qPCR was performed using Bio-Rad iTaq SYBR Green Supermix on a BioRad CFX97 thermocycler and analyzed using the ΔΔCt method. For cultured adipocytes, 18s was used for normalization, and for tissues, the average of *18s* and *B2m* was used for normalization. Primer sequences are listed in Supplementary Table 1.

## Western blotting

Cultured adipocytes or HepG2 cells were harvested in 50 mM Tris-HCl, pH 7.6, 150 mM NaCl, 5 mM EDTA, 1% Triton X-100 supplemented with Halt protease and phosphatase inhibitors (ThermoScientific). The homogenate was spun at $6000 \times g$ for 15 min at 4 °C. The resulting supernatant was considered the "Triton X-100 soluble fraction," and the pellet considered the "Triton X-100 insoluble fraction." To solubilize the pellet, 2.5X Laemmli buffer with β-mercaptoethanol was added. For adipose tissue, fat pads or explants were suspended in 50 mM Tris-HCl, pH 7.6, 150 mM NaCl, 5 mM EDTA with Halt protease and phosphatase inhibitors and homogenized with the Qiagen TissueLyser. Homogenates were spun at $6000 \times g$ for 15 min at 4 °C and the resulting infranatant, between the pellet and overlying fat cake was collected. Protein concentrations were determined by BCA (Pierce). Proteins were prepared for electrophoresis by adding Laemmli buffer with β-mercaptoethanol and heating to ~90 °C for 10 min. Proteins were resolved by SDS-PAGE and blotted with the following antibodies: anti-Fasn (1:1000, CST #3180), anti-Ucp1 (1:1000, Abcam #10983), anti-p62 (1:1000, CST #23214), anti-p62 (1:1000, R&D #MAB8028), anti-alpha-tubulin (1:1000, Sigma #T5168), anti-LC3B (1:1000, CST #83506), anti-LC3A/B (1:1000, CST #12741), anti-Gapdh (1:1000, CST #8884), anti-malonylated lysines (1:1000, PTM Biolabs #901), anti-vinculin (1:1000, CST #18799), anti-Gabarap (1:1000, CST #13733), anti-ubiquitin (1:1000, Proteintech #10201-2-AP), anti-Lamp1 (1:1000, BD Biosciences #553792), anti-Cathepsin B (1:1000, Proteintech #12216-1-AP), anti-β-actin (1:1000, Sigma #A5316).

## Histology

Freshly dissected adipose tissue pieces were fixed in 4% paraformaldehyde overnight and embedded in paraffin. Sections were stained with H&E and anti-p62 (1:500, CST #23214) at the UMass Chan Medical School Morphology Core. Images were taken with a Leica DM2500 LED microscope equipped with a Leica MC170 HD camera.

## Immunofluorescence

Preadipocytes were grown on glass coverslips and differentiated as described. On day 6 or 7, differentiated adipocytes were fixed in 4%

paraformaldehyde for 10 min at room temperature, washed, and permeabilized with ice-cold 100% methanol for 10 min. The cells were incubated with blocking solution containing 2% normal goal serum, 1% BSA, 0.1% Triton X-100, and 0.05% Tween 20. Primary antibodies were diluted in blocking solution and incubated overnight at 4 °C. Secondary antibodies were diluted in blocking solution and incubated at room temperature for 1 hour. Nuclei were stained with 1 μg/ml DAPI (Invitrogen), and coverslips were mounted on slides with Prolong Glass mountant (Invitrogen).

Images were taken with a Leica TCS SP8 confocal microscope. Colocalization analysis was performed with the JACoP plugin in ImageJ. Antibodies: rabbit anti-Fasn (5 ug/mL, Abcam #ab22759); rabbit anti-p62 (1:500, CST #23214); mouse anti-p62 (8 μg/mL, R&D #MAB8028); rabbit anti-LC3A/B (1:200, CST #12741); mouse anti-LC3B (1:200, CST #83506); Alexa Fluor-594 goat anti-rabbit (1:500) and Alexa Fluor-488 goat anti-mouse (1:500, Invitrogen).

For the Magic Red Cathepsin B assay, differentiated WT adipocytes were grown on coverslips and treated with 100 nM TVB3664 or vehicle. After 48 hours, the adipocytes were either fed with regular growth media or starved with HBSS and loaded with Magic Red Cathepsin B substrate (#937, Immunochemistry Technologies) for ~40 min according to the manufacturer's instructions. The adipocytes were washed and fixed in 4% paraformaldehyde for 15 min. Nuclei were stained with DAPI, and coverslips were mounted on slides with Prolong Glass mountant, as described above. Images were taken with the Leica TCS SP8 confocal microscope.

## Lipidomics

Lipid species were analyzed using multidimensional mass spectrometry-based shotgun lipidomic analysis[62]. In brief, homogenates of adipocytic samples containing 0.2 mg of protein as determined by the Pierce BCA assay were accurately transferred to disposable glass culture test tubes. A pre-mixture of lipid internal standards (IS) was added prior to conducting lipid extraction for quantification of the targeted lipid species. Lipid extraction was performed using a modified Bligh and Dyer procedure[63], and each lipid extract was reconstituted in chloroform:methanol (1:1, v-v) at a volume of 500 μL/mg protein.

For shotgun lipidomics, individual lipid extract was further diluted to a final concentration of ~500 fmol total lipids per μL. Mass spectrometric analysis was performed on a triple quadrupole mass spectrometer (TSQ Altis, Thermo Fisher Scientific, San Jose, CA) and a Q Exactive mass spectrometer (Thermo Scientific, San Jose, CA), both of which were equipped with an automated nanospray device (TriVersa NanoMate, Advion Bioscience Ltd., Ithaca, NY) as described[64]. Full and tandem MS scans were automatically acquired by a customized sequence subroutine operated under Xcalibur software (Thermo Fisher 4.2.47). Identification and quantification of lipid species were performed using an automated software program[65,66]. Data processing (e.g., ion peak selection, baseline correction, data transfer, peak intensity comparison and quantitation) was performed as described[66]. The results were normalized to the protein content (nmol or pmol lipid/mg protein). The full lipidomics dataset can be found in Supplementary Data 1.

## Acetyl-CoA and malonyl-CoA measurements

Malonyl- and acetyl-CoA were extracted with 0.3 M perchloric acid and analyzed by LC-MS/MS using a method based on a previously published report by[67]. The extracts were spiked with $^{13}C_2$-Acetyl-CoA (Sigma, MO, USA), centrifuged, and filtered through the Millipore Ultrafree-MC 0.1 μm centrifugal filters before being injected onto the Chromolith FastGradient RP-18e HPLC column, 50 × 2 mm (EMD Millipore) and analyzed on a Waters Xevo TQ-S triple quadrupole mass spectrometer coupled to a Waters Acquity UPLC system (Waters, Milford, MA).

## Proteasome activity assay

Proteasome activity was measured using the Proteasome Activity Fluorometric Assay Kit II (#J4120) from UBPBio, according to the manufacturer's instructions. Briefly, lysates from female Fasn[Fl/Fl] and cAdFasnKO subcutaneous adipose tissues were prepared in buffer containing 50 mM Tris-HCl, pH 7.6, 150 mM NaCl, 5 mM EDTA. 30 μg lysates were used for the assay. Samples were run in duplicate for each substrate with and without MG132 proteasome inhibition. Data presented are the rates without MG132 minus the MG132-inhibited rate.

## Reporting summary

Further information on research design is available in the Nature Portfolio Reporting Summary linked to this article.

## Data availability

Source data are provided with this paper. Uncropped scans of the blots are included in the Source data file. Lipidomics data are provided in Supplementary Data 1. Figure 2a was made under a license with BioRender. Source data are provided with this paper.

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

## Acknowledgements

We thank all members of the Czech laboratory and Drs. Eric Baehrecke and Guangyan Miao for helpful discussions and feedback. The electron microscopy in this project was supported by Award Numbers S10OD025113-01 and S10OD021580 from the National Center for Research Resources. Funding was also provided by grants from the National Institute for Diabetes and Digestive and Kidney Diseases (DK116056 and DK030898 to M.P.C. and DK124723 to C.B.N.). The Functional Lipidomics Core at the Barshop Institute for Longevity and Aging Studies is partially supported by center grants from National Institute on Aging (P30 AG013319 and P30 AG044271). The content is solely the responsibility of the authors and does not necessarily represent the official views of the National Center for Research Resources or the National Institutes of Health.

## Author contributions

L.A.R., A.G., and M.P.C. conceived the study and designed the experiments. L.A.R., A.G., F.H., C.D., S.M., N.W., and M.K. performed the experiments. K.R. and G.H. performed the electron microscopy, M.P. and X.H. performed the lipidomics analyses. O.R.I. and C.B.N. performed the acetyl-CoA and malonyl-CoA measurements and collaborated on interpretations of all the data. L.A.R. and M.P.C. wrote the manuscript, which was reviewed and edited by all co-authors.

## Competing interests

The authors declare no competing interests.
