## [Peer Review File · Nature Communications]

De novo lipogenesis fuels adipocyte autophagosome and lysosome membrane dynamicsREVIEWER COMMENTS

Reviewer #1 (Remarks to the Author):

In this manuscript Rowland et al study the role of fatty acid synthase and DNL in mouse adipocytes. The study is predicated on the fact that adipocytes have enormous capacity for DNL yet they direct very little of their glucose consumption toward triglyceride synthesis. In this study they make the surprising observation that FASN KO adipocytes have a defect in autophagy. They see an accumulation of autophagic structures together with increased levels of p62 and LC3. They go on to show that there is a defect in autophagic flux. They predict that this is due to decreased synthesis of key fatty acids that are required for synthesis of autophagic vacuoles and this is why there is a defect in autophagy in FASN KO cells. Consistent with this they also show that cells in which the activity of ACC is blocked display a similar phenotype. Curiously provision of FASN KO cells with fatty acids in the media is insufficient to fully correct the defect in autophagy. They go on to show that FASN is localised to nascent autophagosomes suggesting there is something specific about FASN at this location that determines the fate of FAs. This is a very interesting study that sheds new light on fat cell biology and on autophagy regulation. The majority of the data in the paper is very solid and well performed. I have just a few general questions:

1. Do the authors have information about how autophagy is regulated in other non-adipose cells? I assume that FASN is widely expressed but does it play a similar role in all cells?
2. It seems counterintuitive that DNL would play a role in autophagy as these processes almost seem mutually exclusive? Please discuss.
3. If there is such a key defect in fat cell autophagy in FASN KO mice do the authors have any evidence for a defect downstream of this autophagy block. Is there a proteostasis defect of some kind? Do these cells accumulate damaged proteins or lipids? This would strengthen the study considerably.
4. It is my understanding that others have reported the increase in protein malonylation in FASN restricted cells? <https://pubs.acs.org/doi/full/10.1021/acs.jproteome.6b00264>

<https://www.sciencedirect.com/science/article/pii/S1550413118304625>

This should be acknowledged and if this is not such a major finding may be these data could be shifted to supplementary particularly since this does not seem to play a role in the phenotype of interest.

5. The in vivo data did not appear to be as strong as the in vitro data. For example increased autophagy in fasted animals was not immediately evident? If anything the autophagy defect in FASN Mice was more evident in fed mice. Please comment.

Reviewer #2 (Remarks to the Author):

The contribution of DNL in phagophore formation and autophagosome formation has been documented, yet the role of DNL in autophagy flux is poorly understood. Using Fasn KO mice and primary adipocytes, Rowland et al elucidated the various aspects of DNL regulation of autophagy. The authors found that cAdFasnKO adipocytes display impaired autophagic degradation with little effect on autophagy activation. More interestingly, it was shown that endogenous fatty acid synthesis directly supplies lipids to autophagosome membranes and is required for effective autophagic flux. Several pieces of nice data demonstrate a novel mechanism underneath the involvement of DNL produced lipids in autophagic degradation, and the overall study provides new insight into autophagy regulation in adipocytes.

1. Does impaired autophagosome maturation and fusion explain the beneficial effects of fasn in vivo as the same group published recently?
2. Depletion of fasn altered lipid droplet size and number. It's unclear whether decreased autophagic flux is secondary to fewer and smaller lipid droplets in adipocytes?
3. Can PI treatment rescue the reduced autophagic flux in fasn KO adipocytes?
4. Phosphoinositides (PI) is critical for lysosome function and homeostasis. Did Lack of fasn has impact on lysosome biogenesis?

Reviewer #3 (Remarks to the Author):

The manuscript by Rowland et al reported a novel role of adipocyte FASN in autophagosome membrane dynamics. Using an adipocyte-specific FASN knockout mouse model, they observed that FASN deficiency in vitro and in vivo led to autophagosome accumulation and severely compromised degradation of the autophagic substrate p62. They further showed that the autophagy impairment appeared to occur at the level of autophagosome-lysosome fusion, and in a lipidomic analysis of adipocytes, loss of FASN results in significant changes of a number of membrane phospholipids, among which phosphoinositides were attributed to the impairment of autophagosome maturation and fusion. In summary, this work shed some new light into the roles of adipocyte DNL in regulating adipose tissue functions, and potentially whole body metabolic homeostasis.

Although the overall conclusions for this manuscript are interesting and likely have significant impact on our understanding of DNL in adipose tissue function, there are some major concerns regarding this manuscript. First of all, the major phenotypic changes (such as the metabolic parameters) in these FASN KO mice with normal chow or HF diet were not included. Are any of these metabolic alterations

associated with the impaired DNL in the adipose tissue, especially due to the impairment of autophagy? Secondly, although FASN is critical for de novo fatty acid synthesis, the authors did not provide unequivocal evidence for the changes of FA synthesis (they only used ratios of C16/C18, Fig 2F) to the autophagic impairment. Data from metabolic flux analysis using ¹³C labeled glucose (or acetate) in vitro or in vivo may consolidate this claim. Thirdly, lipidomic analysis showed changes in multiple phospholipids, tracking these changes by labeled palmitate or glucose may help to dissect the roles of FA from DNL or circulation in modulating adipocyte autophagy.

Below are minor concerns:

1. The authors used ACC inhibitor in this study, why not use FASN inhibitor?
2. In Figure 2C, with ACCi inhibition, levels of protein malonylation in adipocytes of FASN-KO group were still more than that of WT group, does it mean that the increased malonylation of proteins is not entirely serves as a sink for excess malonyl-CoA and other mechanisms likely play a role?
3. Fig. 3G-H showed that FASN colocalized with nascent autophagosomes. Is it possible to observe the dynamic changes of autophagosomes by imaging method?
4. FASN KO leads to beiging of WAT (Figure 4C). However, cardiolipin (CL) and LCL levels were significantly decreased in cAdFasnKO (Figure 5A).
5. The authors attributed the impaired autophagy in FASN KO to the decreased levels of PI (Figure 5A). Further studies on specific phosphatidylinositol species altered by FASN deficiency in vitro and in vivo showed that PI(16:0-18:2), PI(16:1-20:4) and PI(16:0-20:4) were significantly decreased, and PI(18:0-20:4) was significantly increased. These results suggest that PI species may play different roles in autophagy. Since supplementing exogenous FFA cannot completely restore autophagy flux in adipocytes, can specific PI be supplemented to see whether autophagy function is restored?

Response to Referees:

We thank the referees for careful review of our manuscript, and the favorable reviews. We also appreciate the thoughtful and helpful comments and the constructive criticisms that have greatly helped us in improving the manuscript and enhancing the impact of this study. We feel this revised paper will very significantly advance our understanding of adipocyte biology and systemic metabolic regulation. We have revised the text according to the reviewers' comments and have generated much new data now presented in several new multi-panel Supplementary Figures in response to their queries, and we respond to the comments as follows:

REVIEWER COMMENTS

Reviewer #1 (Remarks to the Author):

In this manuscript Rowland et al study the role of fatty acid synthase and DNL in mouse adipocytes. The study is predicated on the fact that adipocytes have enormous capacity for DNL yet they direct very little of their glucose consumption toward triglyceride synthesis. In this study they make the surprising observation that FASN KO adipocytes have a defect in autophagy. They see an accumulation of autophagic structures together with increased levels of p62 and LC3. They go on to show that there is a defect in autophagic flux. They predict that this is due to decreased synthesis of key fatty acids that are required for synthesis of autophagic vacuoles and this is why there is a defect in autophagy in FASN KO cells. Consistent with this they also show that cells in which the activity of ACC is blocked display a similar phenotype. Curiously provision of FASN KO cells with fatty acids in the media is insufficient to fully correct the defect in autophagy. They go on to show that FASN is localised to nascent autophagosomes suggesting there is something specific about FASN at this location that determines the fate of FAs. This is a very interesting study that sheds new light on fat cell biology and on autophagy regulation. The majority of the data in the paper is very solid and well performed. I have just a few general questions:

We thank the reviewer for these supportive comments, especially the view that the paper sheds new light on fat cell biology and autophagy regulation and is mostly very solid and well performed.

1. Do the authors have information about how autophagy is regulated in other non-adipose cells? I assume that FASN is widely expressed but does it play a similar role in all cells?

We thank the reviewer for this important question and the reviewer is correct that Fasn is indeed expressed in most cell types, and thus potentially expected to contribute to autophagy in other cell types. We have performed additional experiments to test this idea and include the new data that supports this concept. Thus, inhibition of Fasn in HepG2 cells for example using a highly specific, potent chemical inhibitor of Fasn (TVB-3664), resulted in a similar phenotype to Fasn KO in adipocytes. Shown in new Supplemental Figure 7, inhibition of Fasn led to increased LC3II lipidation, soluble p62 levels, and insoluble p62. An autophagy flux assay, or LC3II turnover

assay revealed the autophagosome degradation ratio was reduced by Fasn inhibition, similar to what we found with adipocyte FasnKO. Interestingly, the autophagosome synthesis ratio was also significantly reduced by Fasn inhibition in HepG2 cells, which we did not find with adipocyte FasnKO. These data are consistent with a role for Fasn to contribute directly to autophagosome membrane formation in addition to degradation. As we hypothesize in the text, the role of Fasn on autophagosome formation is masked in adipocytes due to compensation by stored lipids. However, in cell types that are not lipid-loaded the effect of Fasn on autophagosome formation can be observed as shown here. We describe this new data with comments in the revised text at the end of the Results section and in the Discussion section.

2. It seems counterintuitive that DNL would play a role in autophagy as these processes almost seem mutually exclusive? Please discuss.

This is a great point, and we have addressed it in the Discussion section of the revised paper. While activation of an anabolic process (DNL) to complete a catabolic process (autophagy) does seem counterintuitive, it's believed that synthesizing autophagosome membranes de novo rather than repurposing old membranes is actually more energetically favorable. The protein and lipid composition of autophagosome membranes, like all cellular membranes, must be tightly controlled to ensure proper function, thus repurposing old membranes would require both lipid and protein remodeling, which would be more energetically costly than de novo synthesis.

3. If there is such a key defect in fat cell autophagy in FASN KO mice do the authors have any evidence for a defect downstream of this autophagy block. Is there a proteostasis defect of some kind? Do these cells accumulate damaged proteins or lipids? This would strengthen the study considerably.

This is a great point, and we thank the reviewer for this comment. We agree this information would contribute meaningfully to the study. We have addressed this issue and included these data in the manuscript as new Supplemental Figure 4. Included in the new text of the Results section: "The autophagic degradation system and the ubiquitin-proteasome system are the major pathways for protein degradation and maintenance of proteostasis, thus blockade of either arm can lead to an accumulation of misfolded and damaged proteins. In addition, crosstalk between these systems has been established, illustrated by the fact that impairment of proteasomal degradation can activate autophagy as a compensatory mechanism, though it's not clear if the alternative is true. To determine if general proteostasis was altered by Fasn deletion or inhibition, we measured total ubiquitinated protein levels. Inhibition of Fasn in cultured adipocytes using the potent inhibitor, TVB-3664, increased soluble p62 and lipidated and unlipidated LC3B, as found with FasnKO adipocytes, though total ubiquitinated proteins were not affected (Figure S4A). Treatment with the proteasomal inhibitor MG132 increased protein ubiquitination similarly in vehicle and Fasn-inhibited adipocytes (Figure S4A). Western

blotting for the Triton X-100 insoluble fraction of these lysates showed an increase in insoluble p62 with Fasn inhibition, as expected, accompanied by a mild increase in protein ubiquitination (Figure S4B). MG132 treatment, however, dramatically increased protein ubiquitination in the insoluble fraction in both groups (Figure S4B). p62 contains a ubiquitin binding domain and mediates the turnover of some ubiquitinated proteins, thus reduced autophagic turnover by Fasn inhibition prevents the degradation of p62-bound ubiquitinated cargo. Compared to the accumulation of ubiquitinated proteins by proteasomal inhibition (MG132); however, the increase with Fasn inhibition is minor and found only in the insoluble fraction. Moreover, western blotting for total ubiquitinated proteins in WAT lysates from cAdFasnKO mice showed no change (Figure S4C).

Lastly, to ensure proteasomal activity was not altered by Fasn KO, we performed a proteasomal activity assay on the WAT samples shown in Figure S4C. We found no differences in activity with any proteasomal substrate (Figure S4D). These findings suggest that while autophagic protein degradation is impaired by Fasn deficiency, general proteostasis is not significantly disrupted. We have added text to the Results section describing the new results in Figure S4. Overall, we found no major defects in general proteostasis in Fasn deficient adipocytes or adipose tissues.”

4. It is my understanding that others have reported the increase in protein malonylation in FASN restricted cells? <https://pubs.acs.org/doi/full/10.1021/acs.jproteome.6b00264>
<https://www.sciencedirect.com/science/article/pii/S1550413118304625>

This should be acknowledged and if this is not such a major finding may be these data could be shifted to supplementary particularly since this does not seem to play a role in the phenotype of interest.

The reviewer is correct that it has been previously reported that blockage of Fasn increases protein malonylation, and we have appropriately referenced these papers in the text. The protein malonylation data included in this manuscript is not meant to be a “discovery.” Since the elevation of protein malonylation with FasnKO is known, and another protein acylation, namely acetylation, is one of the major regulators of autophagy, we feel that addressing the role of protein malonylation in this manuscript was important to do.

5. The in vivo data did not appear to be as strong as the in vitro data. For example increased autophagy in fasted animals was not immediately evident? If anything the autophagy defect in FASN Mice was more evident in fed mice. Please comment.

The reviewer is correct in noting the in vivo effects are less pronounced than the in vitro KO of Fasn. We believe there are a couple of reasons for this. Firstly, while adipocytes make up the largest portion of the volume of adipose tissue, they only account for ~20% of the total cells (Eto et al. Characterization of structure and cellular components of aspirated and excised

adipose tissue. *Plast Reconstr Surg.* 2009 PMID: 19935292). For this reason, the phenotype of adipocytes *in vivo* can be masked by other cell types present in the adipose tissue. This is evident in our study when comparing western blots on whole adipose tissue vs. isolated adipocytes. Shown Figure 4B, changes in LC3 are only consistently evident when isolated adipocytes are analyzed due to a large contribution of LC3 signal from the stromal vascular fraction (SVF), where *Fasn* is not deleted.

Secondly, *in vivo*, adipocytes receive not only more lipids but also a much more diverse set of lipids from the circulation (derived from the diet and liver) compared to *in vitro* cultured adipocytes. Therefore, we believe this may *slightly* offset the effect of *Fasn*KO on autophagy *in vivo*. However, as shown in our *in vitro* fatty acid supplementation studies, exogenous fatty acids are not sufficient to fully restore autophagic function. We find it remarkable though, that despite the presence of circulating fatty acids and full triglyceride stores, adipocyte *Fasn*KO *in vivo* still suppresses autophagic flux, supporting our claim that localized fatty acid synthesis is essential. The above two points are mentioned briefly with the data in the Results section.

Reviewer #2 (Remarks to the Author):

The contribution of DNL in phagophore formation and autophagosome formation has been documented, yet the role of DNL in autophagy flux is poorly understood. Using *Fasn* KO mice and primary adipocytes, Rowland et al elucidated the various aspects of DNL regulation of autophagy. The authors found that cAd*Fasn*KO adipocytes display impaired autophagic degradation with little effect on autophagy activation. More interestingly, it was shown that endogenous fatty acid synthesis directly supplies lipids to autophagosome membranes and is required for effective autophagic flux. Several pieces of nice data demonstrate a novel mechanism underneath the involvement of DNL produced lipids in autophagic degradation, and the overall study provides new insight into autophagy regulation in adipocytes.

We thank the reviewer for the favorable comments and the view that the paper provides nice data demonstrating a novel mechanism involving DNL in autophagic degradation and new insights into autophagy regulation in adipocytes.

1. Does impaired autophagosome maturation and fusion explain the beneficial effects of *fasn* *in vivo* as the same group published recently?

We agree that is an important issue (both Reviewers #2 and #3 posed similar questions), although not the direct point of the study at this stage. This question is most difficult to answer regarding the physiological effects of *Fasn*KO-mediated inhibition of autophagy. Optimally, an experiment to selectively relieve the autophagy inhibition caused by *Fasn*KO *in vivo* would answer this question; however, there are several hurdles which preclude us from addressing this issue in this manuscript, as the experiments to really solve this question are complex and will require major new methods. Because *Fasn*KO blocks autophagy at a late stage, i.e. autophagosome-lysosome fusion/degradation, activating autophagy using known methods (Rapamycin, Tat-Beclin peptide, genetic manipulations), which activate at early stages, would

not be a successful strategy. In addition, the effects of Fasn on autophagy appear to be local. Thus, an ideal experiment would restore Fasn function specifically on or near the autophagy machinery, and this manipulation would need to be done in vivo to address the physiological effects. While we agree these are very important questions, they are the focus of ongoing studies in our lab which we view as long term. Since adipose browning occurs in vivo, the specific manipulations would have to be achieved in vivo, an extremely difficult task. The metabolic benefits of adipose browning and enhanced systemic glucose tolerance are not addressed in this manuscript (only referenced), and we feel are clearly beyond the scope of the present manuscript based on long term experiments that will be necessary to sort out this issue.

2. Depletion of fasn altered lipid droplet size and number. It's unclear whether decreased autophagic flux is secondary to fewer and smaller lipid droplets in adipocytes?

While in vitro differentiated FasnKO adipocytes are quite lipid depleted as the reviewer states, FasnKO adipocytes in vivo are not, yet autophagic dysfunction is still evident. Moreover, our experiments using the ACC inhibitor (Figure 2D) showed that lipid droplets were still abundant while autophagy was inhibited. We have also used an inhibitor of Fasn (TVB3664), which acutely inhibits Fasn and autophagy prior to loss of lipid droplets, with similar results (see Figure below that shows lipid droplets in such a study). Lastly, inhibition of Fasn in non-adipocytes (HepG2 cells) also resulted in autophagy inhibition (new Supplemental Figure 7), indicating diminished lipid droplets do not seem to be the cause of autophagic dysfunction.

Differentiated wild-type adipocytes were treated with 100nM TVB3664 (Fasn inhibitor) or vehicle for 48 hours. Scale bars = 20 μ m

3. Can PI treatment rescue the reduced autophagic flux in fasn KO adipocytes?

As the reviewer has suggested, we performed new experiments to test this idea. We treated adipocytes with the Fasn inhibitor, TVB3664, and concomitantly added varying concentrations of PI. We used a combination of bovine liver and soy PI (Avanti lipids) to ensure full coverage of the PI species changed by Fasn KO. Shown in the figure below, we did not observe a rescue of the autophagy defect caused by Fasn inhibition by PI. At PI concentrations above 80 μ M some toxicity was observed, indicated by increased cell death and morphological changes, which may

explain the changes in LC3B and p62 found in vehicle-treated cells at these higher concentrations. Despite these negative data, they don't preclude a critical role for phosphoinositides in the phenotype of Fasn-deficient cells. We surmise that like palmitate/oleate supplementation of FasnKO adipocytes (Figure 3A), autophagy inhibition could not be sufficiently rescued because of the local requirement for fatty acid synthesis. In other words, the PI supplementation may not have an effect because the PI could not be delivered precisely to the sites where it was needed, i.e. autophagosomes and lysosomes.

Differentiated wild-type adipocytes were treated with 100nM TVB3664 (Fasn inhibitor) or vehicle for 48 hours. At the same time, 0-200uM phosphatidylinositol (derived from bovine liver and soy) was added. Western blots for soluble p62, LC3B, and tubulin (top) and insoluble p62 (bottom) are shown.

4. Phosphoinositides (PI) is critical for lysosome function and homeostasis. Did Lack of fasn has impact on lysosome biogenesis?

The reviewer makes an excellent point, and we performed new experiments to explore this important possibility. We have included these new data in the manuscript (new Supplemental Figure 6). We found that the lysosomal membrane marker, Lamp1, was not changed by FasnKO or inhibition, suggesting normal lysosome abundance. However, lysosomal activity measured by Magic Red staining was impaired by Fasn inhibition. In accordance, levels of the lysosomal protease cathepsin B were significantly reduced by Fasn inhibition. The text describing the new data in Figure S6 is incorporated into the Results section.

While this manuscript was being prepared, a number of papers were published further detailing the events of lysosome reformation during autophagy, and in particular, the important role of phosphoinositides in this process. We hypothesize that the lysosomal impairment we observe by Fasn inhibition is due to a defect in autophagic lysosome reformation as a result of an inability to properly supply membrane components, i.e. phosphoinositides and others, for this process.

Reviewer #3 (Remarks to the Author):

The manuscript by Rowland et al reported a novel role of adipocyte FASN in autophagosome membrane dynamics. Using an adipocyte-specific FASN knockout mouse model, they observed that FASN deficiency in vitro and in vivo led to autophagosome accumulation and severely compromised degradation of the autophagic substrate p62. They further showed that the autophagy impairment appeared to occur at the level of autophagosome-lysosome fusion, and in a lipidomic analysis of adipocytes, loss of FASN results in significant changes of a number of membrane phospholipids, among which phosphoinositides were attributed to the impairment of autophagosome maturation and fusion. In summary, this work shed some new light into the roles of adipocyte DNL in regulating adipose tissue functions, and potentially whole body metabolic homeostasis.

Although the overall conclusions for this manuscript are interesting and likely have significant impact on our understanding of DNL in adipose tissue function, there are some major concerns regarding this manuscript.

We thank the reviewer for the comments that the overall conclusions of our manuscript are interesting and likely have significant impact on understanding DNL in adipose tissue function.

First of all, the major phenotypic changes (such as the metabolic parameters) in these FASN KO mice with normal chow or HF diet were not included. Are any of these metabolic alterations associated with the impaired DNL in the adipose tissue, especially due to the impairment of autophagy?

We agree that is an important issue (both Reviewer #2 and #3 posed similar questions), although not the direct point of the study at this stage. This question is most difficult to answer regarding the physiological effects of FasnKO-mediated inhibition of autophagy. Optimally, an experiment to selectively relieve the autophagy inhibition caused by FasnKO in vivo would answer this question; however, there are several hurdles which preclude us from addressing this issue in this manuscript, as the experiments to really solve this question are complex and will require major new methods. Because FasnKO blocks autophagy at a late stage, i.e. autophagosome-lysosome fusion/degradation, activating autophagy using known methods (Rapamycin, Tat-Beclin peptide, genetic manipulations), which activate at early stages, would not be a successful strategy. In addition, the effects of Fasn on autophagy appear to be local. Thus, an ideal experiment would restore Fasn function specifically on or near the autophagy machinery, and this manipulation would need to be done in vivo to address the physiological effects. While we agree these are very important questions, they are the focus of ongoing studies in our lab which we view as long term. Since adipose browning occurs in vivo, the specific manipulations of Fasn/autophagic/lysosomal membranes would have to be achieved in vivo, an extremely difficult task. The metabolic benefits of adipose browning and enhanced systemic glucose tolerance are not addressed in this manuscript (only referenced), and we feel are clearly beyond the scope of the present manuscript based on long term experiments that will be necessary to sort out this issue.

Secondly, although FASN is critical for de novo fatty acid synthesis, the authors did not provide unequivocal evidence for the changes of FA synthesis (they only used ratios of C16/C18, Fig 2F) to the autophagic impairment. Data from metabolic flux analysis using ¹³C labeled glucose (or acetate) in vitro or in vivo may consolidate this claim.

According to the reviewer's suggestion, we have performed new experiments that directly test whether Fasn KO decrease FA synthesis in our in vitro systems. We have now included in the revised manuscript the new data on fatty acid synthesis measurements from FasnKO adipocytes in vitro (new Supplemental **Figure 1A**). Using ³H-H₂O as a tracer, under both basal and insulin stimulated conditions, fatty acid synthesis is dramatically diminished in FasnKO adipocytes. Thus, the changes we observe in autophagosomes and autophagy dynamics correlate directly with inhibited fatty acid synthesis in adipocytes. We have added text describing this new data in the Results section.

Thirdly, lipidomic analysis showed changes in multiple phospholipids, tracking these changes by labeled palmitate or glucose may help to dissect the roles of FA from DNL or circulation in modulating adipocyte autophagy.

The reviewer raises an interesting point and would provide temporal data on the changes we observe. However, the data would likely provide only correlative data to a very complicated issue since many specific lipids may be involved or indeed only bulk lipid accessibility to autophagic/lysosomal membranes may be needed. In addition, this is an extremely technically challenging issue. A major confounding problem is that other tissues, namely the liver, produce much fatty acid via DNL, which are delivered to adipose tissue; thus, using a tracer such as glucose, would not allow us to determine the contribution of liver-derived fatty acids vs. adipose tissue-derived fatty acids. Without the use of glucose in combination, the use of labeled palmitate would not provide much insight. Also, because in vitro cultured adipocytes are almost entirely reliant on DNL for fatty acids, we also do not feel this would be a proper system to address the question of where the lipids needed are derived. Overall, we feel this is a good, key question that must be tackled in the long term as we explore the detailed molecular mechanisms that are involved in autophagosome/lysosomal membrane dynamics and the steps/sites that DNL is required. Our lab is excited about such studies in the long term, however we feel the initial findings we have made in the present revised manuscript make a major advance that opens a new avenue/field and will foster other labs to join the hunt for the answer to this question posed.

Below are minor concerns:

1. The authors used ACC inhibitor in this study, why not use FASN inhibitor?

In the initial submission of the manuscript, we chose to only use the FasnKO adipocytes for consistency. However, we have performed new experiments based on the reviewer's suggestion. In the newly revised manuscript, we have used a Fasn inhibitor (TVB-3664) as well to our advantage (new Supplemental Figures 4, 6, and 7). As shown, inhibition of Fasn activity

mimics FasnKO, showing it is Fasn catalysis that is key to the results on autophagy. These data more strongly support the central hypothesis of our study and we thank the reviewer for suggesting this approach.

2. In Figure 2C, with ACCi inhibition, levels of protein malonylation in adipocytes of FASN-KO group were still more than that of WT group, does it mean that the increased malonylation of proteins is not entirely serves as a sink for excess malonyl-CoA and other mechanisms likely play a role?

The reviewer is correct in noting that protein malonylation is not merely a “sink” for excess malonyl-CoA. Other studies have noted the occurrence of protein malonylation in various cell types, including bacteria, in the absence of Fasn inhibition and have identified mechanisms for the removal of this post-translational modification (Peng et al. The first identification of lysine malonylation substrates and its regulatory enzyme. *Mol Cell Proteomics*. 2011 PMID: 21908771). For example, glyceraldehyde 3-phosphate dehydrogenase activity has been shown to be inhibited by malonylation, and reversed by Sirt5, which may be relevant physiologically (Nishida et al. SIRT5 Regulates both Cytosolic and Mitochondrial Protein Malonylation with Glycolysis as a Major Target. *Mol Cell*. 2015 PMID: 26073543). Thus, we agree with the reviewer that there are likely normal, physiologic functions of protein malonylation; however, a build-up of malonyl-CoA, such as during Fasn inhibition, can drive increased protein malonylation, either as a protective mechanism or merely as a consequence.

3. Fig. 3G-H showed that FASN colocalized with nascent autophagosomes. Is it possible to observe the dynamic changes of autophagosomes by imaging method?

This would be challenging, as live cell microscopy would likely be the best approach. We have not yet approached this issue by imaging.

4. FASN KO leads to beiging of WAT (Figure 4C). However, cardiolipin (CL) and LCL levels were significantly decreased in cAdFasnKO (Figure 5A).

The reviewer has astutely noted that cardiolipin, which is required for beiging of adipose tissue (Sustarsic et al. Cardiolipin Synthesis in Brown and Beige Fat Mitochondria Is Essential for Systemic Energy Homeostasis. *Cell Metab*. 2018 PMID: 29861389) is decreased, seemingly in disagreement with the beige phenotype of FasnKO adipose tissue. However, we only observe this decrease in FasnKO adipocytes *in vitro*, where we have not observed increased thermogenic gene expression. In contrast, *in vivo*, in FasnKO adipose tissue, where there is increased beiging, i.e. Ucp1 expression, cardiolipin levels are not changed (Figure 5B); therefore, we do not believe cardiolipin levels, *in vivo*, play a role in the phenotype we observe.

5. The authors attributed the impaired autophagy in FASN KO to the decreased levels of PI (Figure 5A). Further studies on specific phosphatidylinositol species altered by FASN deficiency *in vitro* and *in vivo* showed that PI(16:0-18:2), PI(16:1-20:4) and PI(16:0-20:4) were significantly decreased, and PI(18:0-20:4) was significantly increased. These results suggest that PI species

may play different roles in autophagy. Since supplementing exogenous FFA cannot completely restore autophagy flux in adipocytes, can specific PI be supplemented to see whether autophagy function is restored?

The reviewer raises a good question, and a similar question was posed by reviewer #2. In response to both referees, we treated adipocytes with the Fasn inhibitor, TVB3664 and concomitantly added varying concentrations of PI. We used a combination of bovine liver and soy PI (Avanti lipids) to ensure full coverage of the PI species changed by Fasn KO. Shown in the figure below, we did not observe a rescue of the autophagy defect caused by Fasn inhibition by PI. At PI concentrations above 80 μ M some toxicity was observed, indicated by increased cell death and morphological changes, which may explain the changes in LC3B and p62 found in vehicle-treated cells at these higher concentrations. Despite these negative data, they don't preclude a critical role for phosphoinositides in the phenotype of Fasn-deficient cells. We surmise that like palmitate/oleate supplementation of FasnKO adipocytes (Figure 3A), autophagy inhibition could not be sufficiently rescued because of the local requirement for fatty acid synthesis. In other words, the PI supplementation had no effect because the PI could not be delivered precisely to the sites where it was needed, i.e. autophagosomes and lysosomes.

Differentiated wild-type adipocytes were treated with 100nM TVB3664 (Fasn inhibitor) or vehicle for 48 hours. At the same time, 0-200 μ M phosphatidylinositol (derived from bovine liver and soy) was added. Western blots for soluble p62, LC3B, and tubulin (top) and insoluble p62 (bottom) are shown.

REVIEWERS' COMMENTS

Reviewer #1 (Remarks to the Author):

The authors have done a very thorough job of addressing all of my comments. I feel this manuscript has been markedly improved and I have no further comments. I congratulate the authors on an excellent study.

Reviewer #2 (Remarks to the Author):

The authors have adequately and/or experimentally addressed my questions. I have no other concerns.

Reviewer #3 (Remarks to the Author):

The authors adequately addressed all my concerns

REVIEWERS' COMMENTS

Reviewer #1 (Remarks to the Author):

The authors have done a very thorough job of addressing all of my comments. I feel this manuscript has been markedly improved and I have no further comments. I congratulate the authors on an excellent study.

Author Response: We thank Reviewer 1 for their kind remarks and thoughtful reviews.

Reviewer #2 (Remarks to the Author):

The authors have adequately and/or experimentally addressed my questions. I have no other concerns.

Author Response: We thank Reviewer 2 for their thoughtful reviews and comments.

Reviewer #3 (Remarks to the Author):

The authors adequately addressed all my concerns

Author Response: We thank Reviewer 3 for their thoughtful reviews and comments.